# Splatter a Video: Video Gaussian Representation for Versatile Processing

**Yang-Tian Sun**[1*]    **Yi-Hua Huang**[1*]    **Lin Ma**    **Xiaoyang Lyu**[1]    **Yan-Pei Cao**[2]    **Xiaojuan Qi**[1†]

[1]The University of Hong Kong    [2] VAST

## Abstract

Video representation is a long-standing problem that is crucial for various downstream tasks, such as tracking, depth prediction, segmentation, view synthesis, and editing. However, current methods either struggle to model complex motions due to the absence of 3D structure or rely on implicit 3D representations that are ill-suited for manipulation tasks. To address these challenges, we introduce a novel explicit 3D representation—video Gaussian representation—that embeds a video into 3D Gaussians. Our proposed representation models video appearance in a 3D canonical space using explicit Gaussians as proxies and associates each Gaussian with 3D motions for video motion. This approach offers a more intrinsic and explicit representation than layered atlas or volumetric pixel matrices. To obtain such a representation, we distill 2D priors, such as optical flow and depth, from foundation models to regularize learning in this ill-posed setting. Extensive applications demonstrate the versatility of our new video representation. It has been proven effective in numerous video processing tasks, including tracking, consistent video depth and feature refinement, motion and appearance editing, and stereoscopic video generation.

## 1   Introduction

Video processing, which encompasses a variety of tasks such as video editing, can enable numerous applications in fields like social media, filmmaking, and advertising [2, 50]. A video can be viewed as a collection of spatiotemporal pixels. However, processing a video directly in its pixel space, while maintaining temporal consistency, poses challenges due to the inherent complexities associated with appearance, motion, occlusions, and noise in the video data [14, 27]. Consequently, a robust video representation capable of abstracting and disentangling appearance and motion is crucial for facilitating various applications and overcoming these challenges.

Existing research on video representation for processing has primarily focused on 2D/2.5D techniques, employing methods such as optical flow and tracking to associate pixels across frames [47, 14, 58]. These approaches often involve learning a canonical image [12, 33, 49, 30] or a layered atlas with persistent motion patterns [14, 24, 4, 9] to facilitate editing and then use optical flow or tracks to propagate edits throughout a video. The most recent work [33] utilizes hash grids combined with implicit functions to embed a video into a learned canonical image for appearance and a deformation field for motion. Despite achieving promising results in appearance editing tasks, these methods struggle to handle occlusions of objects (see Fig. 3), leading to erroneous propagation. Although layered 2.5D representation [14, 24, 4, 9] can mitigate this issue, they still face challenges with complex self-occlusions within a layer. Moreover, these techniques have limited or no capability in

---

*Equal Contribution †Corresponding Author
Project page: `https://sunyangtian.github.io/spatter_a_video_web/`.

addressing processing tasks that require 3D information, such as video representation with complex occlusions, consistent depth prediction, and stereoscopic video generation.

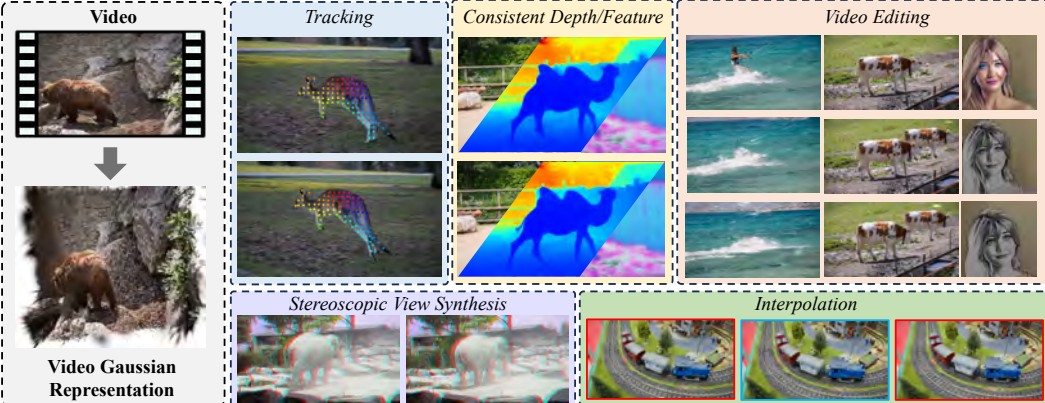

Figure 1: We propose an approach to convert a video into a Video Gaussian Representation (VGR), which can be used for versatile video processing tasks conveniently.

Drawing inspiration from the fact that a video is essentially a projection of the dynamic 3D world onto the 2D image plane at different moments, we pose the question: *is it possible to represent a video in its intrinsic 3D form?* By doing so, we could potentially bypass the limitations of 2D representations, such as occlusions, reduce the complexity of motion modeling, and support processing tasks that require 3D information. Recent work [48] has explored 3D representations, which employ an implicit radiance field to model a canonical 3D space and leverage a bi-directional mapping network for associating 2D pixels with 3D representations. While this approach demonstrates promising performance in dense tracking, it falls short in faithfully representing video appearance, making it incapable of performing video processing tasks that require generating new videos, such as video editing. Moreover, its implicit nature limits its applicability to a variety of video processing tasks that require explicit content or motion manipulations, such as the removal or addition of objects and adjustments to the motion patterns of objects.

In this paper, we introduce a novel explicit video Gaussian representation (VGR) based on 3D Gaussians [16]. Our core idea revolves around utilizing Gaussians in a canonical 3D space to model video appearance while associating each Gaussian with time-dependent 3D motion attributes to control its locations at different time steps for video motion. This 3D representation can then be employed to process and render videos effectively. The subsequent challenge lies in how to map a video onto such a 3D Gaussian representation. This is inherently difficult due to the loss of essential 3D information during 3D-to-2D projection, as well as the entanglement of motion and appearance in videos. However, recent advancements in large models have facilitated the acquisition of high-quality monocular priors from images and videos, such as optical flow [44, 11] and monocular depth [54, 15, 53]. While these 2D priors may not be perfect, they can serve as regularization for learning through knowledge distillation. Consequently, we propose leveraging these 2D priors in conjunction with our 3D motion regularization for learning. By doing so, we effectively lift 2D information– such as pixels, depth, and optical flow–into a unified and compact 3D representation.

Upon learning, our video Gaussian representation can be used to support versatile video processing tasks, as shown in Fig. 1. Here, we showcase its efficacy in 7 video-processing tasks: Specifically, it can be used to obtain **1)** dense tracking and **2)** improve the consistency of monocular 2D prior across frames, leading to better video depth and feature consistency. Secondly, our representation facilitates a range of video editing tasks, including **3)** geometry editing and **4)** appearance editing. Thirdly, it also proves useful in video interpolation, allowing for **5)** the generation of smooth transitions between frames. Finally, as our representation is inherently 3D, it opens up additional possibilities, such as **6)** novel view synthesis (to a certain extent) and **7)** the creation of stereoscopic videos.

## 2    Related Work

As our method utilizes dynamic 3D Gaussians to represent videos and supports versatile video processing, this section introduces related works on video editing, tracking, and dynamic Gaussian splatting. We briefly cover the most relevant works. For additional references, see Sec.A.7.

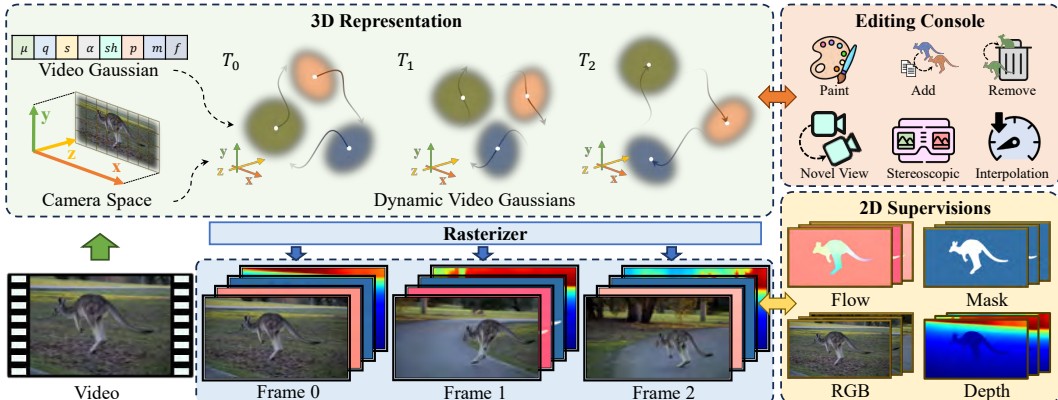

Figure 2: **Pipeline of our approach.** Given a video, we represent its intricate 3D content using video Gaussians in the camera coordinate space. By associating them with motion parameters, we enable video Gaussians to capture the video dynamics. These video Gaussians are supervised by RGB image frames and 2D priors such as optical flow, depth, and label masks. This representation makes it convenient for users to perform various editing tasks on the video.

**Video Editing**  Decomposing videos into layered representations facilitates advanced video editing techniques. Kasten et al. [14] introduced layered neural atlases, enabling efficient video propagation and editing. Further advancements include deformable sprites [58], bi-directional warping fields [9], and innovations in rendering lighting and color details [4]. CoDeF [33] and GenDeF [49] focus on multi-resolution hash grids and shallow MLPs for frame-by-frame deformations. Latent diffusion models [38] and methodologies like ControlVideo [61], MaskINT [28], and VidToMe [20] have also been employed for data-driven video editing.

**Video Tracking**  Video tracking captures physical motion within video sequences. PIPs [8] and TAPIR [6] offer foundational approaches, while CoTracker [13] uses a sliding-window transformer for tracking. OminiMotion [48] and MFT [30] employ neural radiance fields and optical flow fields for dense tracking. State-of-the-art methods like RAFT [44] and FlowFormer [11] provide accurate flow estimations but struggle with long-term correspondences.

**Dynamic Gaussian Splatting**  Gaussian Splatting [16] enhances rendering in radiance fields and has been extended to dynamic scenes [26, 57, 51]. Methods like SC-GS [10] and 3DGStream [43] offer novel approaches for scene dynamics. Our method targets **monocular video representation**, eliminating the need for camera pose estimations and facilitating robust long-term tracking and editing in dynamic scenes.

## 3   3D Gaussian Splatting

Gaussian splatting [16] models 3D scenes using Gaussians learned from multiview images. Each Gaussian, $G$, is defined by a center $\mu$ and a covariance matrix $\Sigma$: $G(x) = \exp\left(-\frac{1}{2}(x-\mu)^T \Sigma^{-1}(x-\mu)\right)$. Here, $\Sigma$ is decomposed into $RSS^T R^T$ for optimization, with $R$ as a rotation matrix parameterized by a quaternion $q$ and $S$ as a scaling matrix parameterized by a vector $s$. Each Gaussian also has an opacity $\alpha$ and spherical harmonic ($\mathcal{SH}$) coefficients $sh$. Then 3D Gaussians can be formulated as: $\mathcal{G} = \{G_j : \mu_j, q_j, s_j, \alpha_j, sh_j\}$. Rendering is done via:

$$C(u) = \sum_{i \in N} T_i \sigma_i \mathcal{SH}(sh_i, v_i), T_i = \Pi_{j=1}^{i-1}(1 - \sigma_j), \tag{1}$$

where $\sigma_i$ is calculated by projecting Gaussian $G_i$ at the rendering pixel and $v$ is the direction from view point to the Gaussian. Optimizing parameters $\{G_j : \mu_j, q_j, s_j, \alpha_j, sh_j\}$ and adjusting densities allows for high-quality, real-time image synthesis. For a more detailed introduction to Gaussian Splatting, please refer to Sec. A.8. We extend 3D Gaussians to represent a video by adding attributes to Gaussians for versatile processing.

# 4 Method

Given a video, our goal is to use 3D Gaussians in a canonical space to represent its appearance and associate Gaussians with 3D motions for video dynamics. To facilitate this mapping, we incorporate 2D priors extracted from existing 2D models and apply 3D motion regularization. This representation allows us to efficiently perform various downstream applications. The pipeline of our method is depicted in Fig. 2. In the following, we elaborate on the video Gaussian representation in Sec.4.1. Then, we discuss the learning objectives and optimization details in Sec.4.2 and Sec. 4.3, respectively.

## 4.1 Video Gaussian Representation

**Camera Coordinate Space** Instead of utilizing an absolute 3D world coordinate system, we opt for the orthographic camera coordinate system to model a video's 3D structure, as demonstrated in Omnimotion [48]. In this space, the video's width, height, and depth correspond to the $X$, $Y$, and $Z$ axes, respectively. This enables us to circumvent the challenges associated with estimating camera poses or disentangling camera motion from scene dynamics, which can be not only time-consuming [40, 41] but also prone to failure in casually captured monocular videos with dynamic objects [34, 59]. By modeling the scene as dynamic 3D Gaussians in the camera coordinate space, we intertwine camera motion with object motion and treat them as the same type of motion, eliminating the need for camera calibration. During the rendering process, the 3D Gaussians in the camera coordinate space are rasterized into images from an identity pose camera. This approach simplifies the representation of dynamics and avoids the challenges of estimating camera pose from monocular casual videos.

**Video Gaussians** Given a video $\mathcal{V} = \{I_1, I_2, \ldots, I_n\}$ consisting of $n$ frames, our video Gaussian representation transforms it into a set of dynamic 3D Gaussians, parameterized as $\mathcal{G} = \{G_1, G_2, \ldots, G_m\}$, to simultaneously represent the appearance and motion dynamics of the video. Each Gaussian is characterized by its position $\mu$, rotation quaternion $q$, scale $s$, spherical harmonics ($\mathcal{SH}$) coefficients of appearance $sh$, and opacity $\alpha$. In addition to these fundamental Gaussian properties for appearance, dynamic attributes $p$, segmentation labels $m$, and image features $f$ from any 2D base models (e.g., DINOv2 [31] and SAM [17]) can also be associated with 3D Gaussians to depict the video's scene content. Consequently, a Gaussian can be expressed as $G = (\mu, q, s, \alpha, sh, p, m, f)$. To learn these properties from a video, we enhance the differentiable 3D Gaussian renderer to render additional attributes beyond simple color, which we denote as $\mathcal{R}(\mu, q, s, \alpha, x)$, where $x$ represents the specific attribute to be rendered. The rendering function $\mathcal{R}$ follows the same procedure as color rendering in the original Gaussian Splatting method [16].

**Gaussian Dynamics** When parameterizing motion, there is a trade-off between incorporating more regularization from motion priors and achieving high fitting capability [46]. In line with recent popular methods [21, 18], we employ a flexible set of hybrid bases comprising polynomials [22] and Fourier series [1] to model smooth 3D trajectories. Specifically, we assign learnable polynomial and Fourier coefficients to each Gaussian, denoted as $p = \{p_p^n\} \cup \{p_{\sin}^l, p_{\cos}^l\}$, respectively. Here, $n$ and $l$ represent the order of coefficients. The position of a Gaussian at time $t$ can then be determined as follows:

$$\mu(t) = \mu_0 + \sum_{n=0}^{N} p_p^n t^n + \sum_{l=0}^{L} (p_{\sin}^l \cos(lt) + p_{\cos}^l \sin(lt)). \tag{2}$$

Polynomial bases $\{t^n\}$ are effective in modeling overall trends and local non-periodic variations in motion trajectories and are widely used in curve representation, such as in Bezier and B-spline curves [32, 7]. Fourier bases $\{\cos(lt), \sin(lt)\}$ offer a frequency domain parameterization of curves, making them suitable for fitting smooth movements [1], and excel in capturing periodic motion components. The combination of these two bases leverages the strengths of both, providing comprehensive modeling, enhanced flexibility and accuracy, reduced overfitting, and robustness to noise. This equips Gaussians with the adaptability to fit various types of trajectories by adjusting the corresponding learnable coefficients. It is important to note that for each Gaussian, the associated parameters $p = \{p_p^n\} \cup \{p_{\sin}^l, p_{\cos}^l\}$ are learned from the video by optimizing the learning objective as described in Sec. 4.3.

## 4.2 2D Monocular Priors and 3D Motion Regularization

Learning video Gaussians in the camera coordinate space to achieve consistency with real-world content using photometric loss is challenging and often ill-posed. There are multiple solutions for video Gaussians to fit the observed 2D projections. For instance, relative depth orders among scene objects can be ambiguous without occlusion cues. Moreover, different Gaussians may sequentially represent the same object, and their motion may not precisely match the object's actual motion. Therefore, regularization is required during the training process.

Thanks to advancements in 2D visual understanding methods, monocular 2D priors such as optical flow [44, 11] and depth estimation [54, 15, 53] are now accessible. Although not perfect, these priors can provide crucial cues to regularize learning. To stabilize our method's training and ensure a real-world consistent solution, we supervise the video Gaussians using priors from the estimated flow obtained from RAFT [44] and the estimated depth derived from Marigold [15].

**Flow Distillation** Optical flow represents the 2D projection of 3D motion. Flow distillation serves to regularize the 2D projections of 3D Gaussian motions. To guarantee that the motion of video Gaussians aligns with the estimated optical flow, we project the 3D motion of Gaussians ($\mu(t_2) - \mu(t_1)$) between frames $t_1$ and $t_2$ onto the 2D image plane and regularize it using the estimated optical flow:

$$\mathcal{L}_{\text{flow}} = \mathbb{E}_{(t_1,t_2)}\left(\|\mathcal{R}(\mu(t_1), q, s, \alpha, \pi(\mu(t_2)) - \pi(\mu(t_1))) - \text{flow}_{t_1 \to t_2}\|_1\right). \quad (3)$$

Here, $\pi$ denotes the projection function that maps camera coordinates to image coordinates after projection, and $\text{flow}_{t_1 \to t_2}$ represents the optical flow estimated by RAFT [44] from $t_1$ to $t_2$. This prior aids video Gaussians in learning the scene flow by ensuring that the 2D projection of their 3D motion on the XY-plane is consistent with the optical flow instead of relying on relative depth changes along the Z-axis to fit frame colors.

**Depth Distillation** Monocular depth estimation provides the per-frame depth of a video. Although these estimates may be inconsistent across long-range frames, they offer valuable cues for regularizing the scene geometry. As a result, we utilize depth maps estimated by Marigold [15] to ensure a reasonable geometry for our video Gaussians. We employ the scale- and shift-trimmed loss proposed in MiDaS [37]:

$$\mathcal{L}_{\text{depth}} = \mathbb{E}_t\left(\|\tau(D^t) - \tau(\hat{D}^t)\|^2\right), \tau(D^t) = (D^t - t(D^t))/\overline{|D^t - t(D^t)|}, t(D^t) = \text{median}(D^t), \quad (4)$$

where $D^t$ is the rendered depth of 3D Gaussians at time $t$, and $\hat{D}^t$ is the corresponding predicted depth. It is worth noting that, thanks to our 3D representation, our approach can, in turn, refine the inconsistent monocular depth estimations and yield consistent depth predictions for a video.

In sum, flow distillation regularizes the projected 3D Gaussian motion on the 2D image plane, corresponding to the X-Y axes in the camera coordinate space. Meanwhile, depth distillation regularizes the relative video Gaussian positions corresponding to the Z-axis in the camera coordinate space. Together, they offer comprehensive 3D supervision and complement each other, effectively regularizing the learning of 3D motion for video Gaussians.

**3D Motion Regularization** In addition to depth and flow distillation, we employ local rigidity regularization to prevent Gaussians from overfitting the rendering targets through non-rigid motions [10, 26]. This approach encourages the 3D motion of individual Gaussians to be as locally rigid as possible [42]. As a result, Gaussians form locally rigid structures, aligning with real-world dynamics. To constrain the local rigidity of a Gaussian $G_i$ from time $t_1$ to $t_2$, we first identify the $K$ nearest neighboring Gaussians $G_k(k \in \mathcal{N}_i)$ using its 3D position at $t_1$. Then, we apply the rigid loss to ensure that the edges between them $(\mu_i(t_1) - \mu_k(t_1))$ adhere to a rigid transformation:

$$\mathcal{L}_{\text{arap}} = \mathbb{E}_{(i,t_1,t_2)}\left(\sum_{k \in \mathcal{N}_i} \|(\mu_i(t_1) - \mu_k(t_1)) - \hat{R}_i(\mu_i(t_2) - \mu_k(t_2))\|^2\right), \quad (5)$$

where $R$ is the estimated rigid rotation transformation given by

$$\hat{R}_i = \arg\min_{R \in \mathbf{SO}(3)} \sum_{k \in \mathcal{N}_i} \|\mu_i(t_1) - \mu_k(t_1)) - R(\mu_i(t_2) - \mu_k(t_2))\|^2. \quad (6)$$

### 4.3 Optimization

In addition to 2D priors and 3D regularization for learning 3D motion and geometry, we also incorporate a color rendering loss for appearance learning. Furthermore, we introduce an optional mask loss to facilitate the separation of background and foreground, which is particularly useful for editing applications.

**Color Rendering Loss** Video Gaussian representation also learns to fit the color of video frames $\{I_{gt}^t\}$ as in novel view synthesis methods [16, 29] with the rendering loss:

$$\mathcal{L}_{\text{render}} = \mathbb{E}_t \left( ||\mathcal{R}(\mu(t), q, s, \alpha, \mathcal{SH}(sh, v)) - I_{gt}^t|| \right). \tag{7}$$

**Mask Loss** Segmentation labels serve as a crucial attribute for pixels, enabling the identification of groups of pixels belonging to foreground objects. In our experiments, we separate pixels into foreground and background components by segmenting each frame and extracting the foreground mask $\mathcal{M}^t$. This mask is subsequently lifted to Gaussian, where it is associated with a learnable label attribute $m \in \{0, 1\}$. The label attributes of Gaussians are supervised by the image segmentation results:

$$\mathcal{L}_{\text{label}} = \mathbb{E}_t \left( ||\mathcal{R}(\mu(t), q, s, \alpha, m) - \mathcal{M}^t||_2^2 \right). \tag{8}$$

With the segmentation label, we can divide Gaussians into different parts and constrain their motion respectively, as shown in Eq. 5. Our approach can also manipulate (remove/duplicate) and edit specific objects in a video, as shown in Fig 7.

**Total Learning Objective** The total learning objective is the weighted sum of all the losses:

$$\mathcal{L} = \lambda_{\text{render}}\mathcal{L}_{\text{render}} + \lambda_{\text{depth}}\mathcal{L}_{\text{depth}} + \lambda_{\text{flow}}\mathcal{L}_{\text{flow}} + \lambda_{\text{arap}}\mathcal{L}_{\text{arap}} + \lambda_{\text{label}}\mathcal{L}_{\text{label}}. \tag{9}$$

**Adaptive Density Control** We initialize the video Gaussians by uniformly sampling points in the camera coordinate space of the first frame, and apply a similar density control strategy as in vanilla Gaussian Splatting [16]. For more details, please refer to Sec. A.1.

## 5 Video Processing Applications

With our video Gaussian representation, we can perform various video processing tasks, including **1)** dense tracking, **2)** consistent depth/feature prediction, **3)** geometry editing, **4)** appearance editing, **5)** frame interpolation, **6)** novel view synthesis, and **7)** stereoscopic video creation. In this section, we detail these applications, highlighting the versatility of video Gaussians.

**Dense Tracking** Since the scene motion is captured by the dynamics of video Gaussians, we can project these dynamics onto the image plane as UV flow and rasterize the attributes as flow maps. This method handles both short and long-frame gaps effectively. The pixel flow map $dU_{t_1 \rightarrow t_2}$ from $t_1$ to $t_2$ is calculated as:

$$dU_{t_1 \rightarrow t_2} = \mathcal{R}(\mu(t_1), q, s, \alpha, \pi(\mu(t_2)) - \pi(\mu(t_1))). \tag{10}$$

The rendered dense flow map provides pixel correspondences, facilitating tracking across frames.

**Consistent Depth/Feature Prediction** Video Gaussians, supervised by monocular depth priors for each frame, conform to a reasonable geometry layout, providing consistent depth predictions across frames. Similarly, other image features can be distilled into video Gaussians; unifying per-frame features into a consistent 3D form. To distill image features (e.g., SAM [17] or DINOv2 [31]), we associate each video Gaussian with a feature attribute $f$ and rasterize them to match the feature map $\{\mathcal{F}_{gt}^t\}$ from 2D models:

$$\mathcal{L}_{\text{feature}} = \mathbb{E}_t \left( ||\mathcal{R}(\mu(t), q, s, \alpha, f) - \mathcal{F}_{gt}^t||_2^2 \right). \tag{11}$$

Optimizing video Gaussians with $\mathcal{L}_{\text{feature}}$ unifies frame-wise 2D features in a 3D form, enabling the rendering of view-consistent feature maps $\{\mathcal{F}_t\}$:

$$\mathcal{F}_t = \mathcal{R}(\mu(t), q, s, \alpha, f). \tag{12}$$

Consistent feature prediction is crucial for applications like video segmentation and re-identification.

CoDeF      Omnimotion      Ours      GT

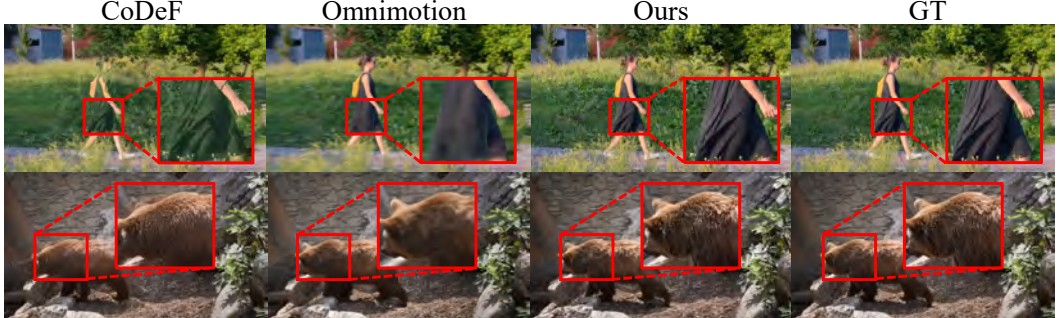

Figure 3: Qualitative comparison of video reconstruction using our method and SOTA methods.

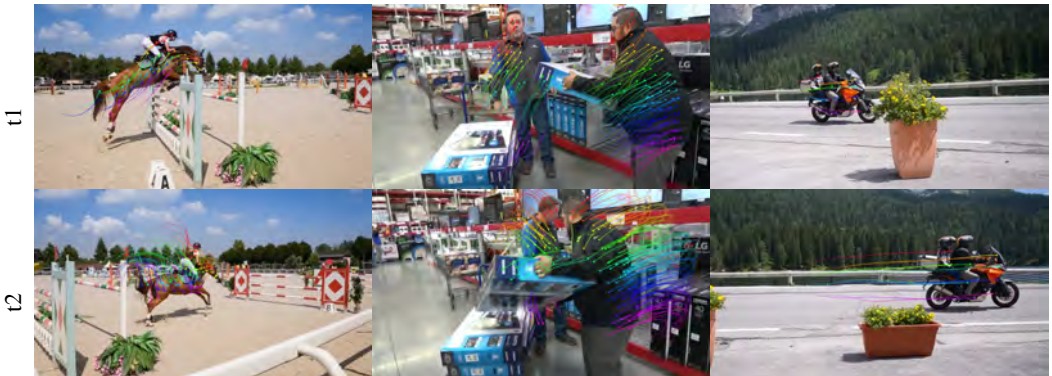

Figure 4: Dense tracking results on diverse complex motion patterns.

**Geometry Editing** In the unified 3D space, geometry editing is straightforward. By distilling segmentation labels into video Gaussians, we can select Gaussians of the target identity and transform their positions $\mu$, quaternions $q$, and scales $s$ for translation, resizing, and rotation. Adjusting their opacities changes the transparency of the edited objects. It also facilitates easy object removal within a video and supports object copying both between and within videos.

**Appearance Editing** Appearance editing with video Gaussians can also be easily achieved. Users can select a specific frame $t$ and perform painting, recoloring, or stylization. We fix all attributes except the $\mathcal{SH}$ coefficients representing Gaussian appearance and optimize them to fit the edited image $I_{\text{edit}}^t$ using:

$$\mathcal{L}_{\text{edit}} = ||\mathcal{R}(\mu(t), q, s, \alpha, \mathcal{SH}(sh, v)) - I_{\text{edit}}^t||_2^2. \tag{13}$$

The edited results can propagate throughout the video, maintaining temporal consistency.

**Frame Interpolation** The learned smooth trajectories of video Gaussians enable interpolation of scene dynamics at any up-sampling rate. Interpolated Gaussians' dynamic attributes can render interpolated video frames. By re-mapping the timestep values $\{t\} \rightarrow \{t'\}$ with an arbitrary continuous function, we can freely adjust the video playback speed.

**Novel View Synthesis** Applying a global rigid transformation $\mathcal{T} \in \mathcal{SE}(3)$ to video Gaussians allows for camera position adjustments. The rendering results of transformed Gaussians $\mathcal{R}(\mathcal{T}(\mu(t)), \mathcal{T}(q), s, \alpha, \mathcal{SH}(sh, v))$ provide synthesized views from different perspectives.

**Stereoscopic Video Creation** Similar to the novel view synthesis application, we can achieve stereoscopic frames by slightly translating video Gaussians horizontally by a fixed distance, representing the interocular distance. This application is crucial in filmmaking and gaming.

## 6 Experiments

**Evaluation** We conducted experiments on the DAVIS dataset [36] as well as some videos used by Omnimotion [48] and CoDeF [33]. Our approach is evaluated based on two criteria: 1) reconstructed video quality and 2) downstream video processing tasks. In addition to general video representation methods Deformable Sprites [58], Omnimotion [48] and CoDeF [33], we also compare with dynamic

Table 1: Comparison with existing methods on Tap-Vid benchmark (DAVIS).

| Methods | PSNR↑ | SSIM↑ | LPIPS↓ | AJ↑ | $\delta^x_{avg}$ ↑ | OA↑ | TC↓ | Training Time | GPU Memory | FPS↑ |
|---|---|---|---|---|---|---|---|---|---|---|
| 4DGS [51] | 18.12 | 0.5735 | 0.5130 | 5.1 | 10.2 | 75.45 | 8.11 | ~40 mins | 10G | 145.8 |
| RoDynRF [23] | 24.79 | 0.723 | 0.394 | / | / | / | / | >24 hours | 24G | >1min |
| Deformable Sprites [58] | 22.83 | 0.6983 | 0.3014 | 20.6 | 32.9 | 69.7 | 2.07 | ~30 mins | 24G | 1.6 |
| Omnimotion [48] | 24.11 | 0.7145 | 0.3713 | **51.7** | **67.5** | **85.3** | **0.74** | >24 hours | 24G | >1min |
| CoDeF [33] | 26.17 | 0.8160 | 0.2905 | 7.6 | 13.7 | 78.0 | 7.56 | ~30 mins | 10G | 8.8 |
| Ours | **28.63** | **0.8373** | **0.2283** | 41.9 | 57.7 | 79.2 | 1.82 | ~30 mins | 10G | 149 |

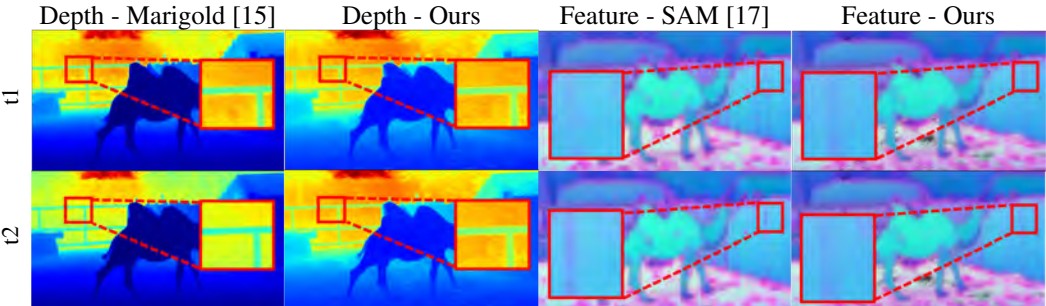

Figure 5: Qualitative comparison of video depth and features generated by our method and SOTA single-frame estimation methods. Our method yields more consistent estimations.

NeRF/3DGS methods, namely 4DGS [51] and RoDynRF [23]. Note that for 4DGS, we estimate camera poses using monocular depth estimation method Unidepth [35] and DROID-SLAM [45]. Despite these efforts, the performance remains unsatisfactory, further highlighting the challenges of accurate camera pose estimation in causal dynamic videos. In contrast, our approach demonstrates the capability to handle more complex motions and achieves significantly higher reconstruction quality. For downstream tasks, our method also shows comparable performance to those specifically designed for these tasks.

**Video Reconstruction** To demonstrate our method's fitting ability for casual videos, we compare it with Omnimotion [48] and CoDeF [33]. Omnimotion tends to render blurred results due to the smooth bias of the MLP when modeling the canonical space, while CoDeF struggles with complex motions due to the limited representation ability of the 2D canonical image. We report the rendering quality metrics and visualizations on the Tap-Vid DAVIS dataset [36] in Table 1 and Figure 3. More comparison with RyDynRF [23] and 4DGS [51] are provided in the supplementary materials.

### 6.1 Video Processing Applications

**Dense Tracking.** Our approach enables dense tracking by projecting the dynamics of Gaussians onto 2D image planes to obtain correspondences. Tracking results are visualized in Fig. 4 and evaluated in Table 1. Despite Omnimotion's specialization in tracking, our approach supports a wider array of video processing tasks with higher computational and training efficiencies. It achieves comparable results with better reconstruction quality using fewer resources. Tracking performance comparisons with similar-cost methods (CoDeF / 4DGS) are shown in the supplementary materials 11, highlighting our superior outcomes.

**Consistent Depth / Feature Generation.** We present the results of consistent video depth and features (using SAM [17]) in Fig. 5, compared to per-frame prediction. Due to the unified 3D representation of video frames, the predicted depth and features exhibit significantly better consistency than those obtained from monocular predictions. We also evaluate the effectiveness of consistent SAM feature in the segmentation task in the supplementary materials 15. We recommend that readers watch the supplemental videos for better illustrations.

**Geometry Editing** By manipulating the Gaussians associated with specific labels, we can achieve geometric editing of target identities, as demonstrated in Fig. 7. By deleting foreground Gaussians, we can remove foreground elements and render a clean background. Our approach also supports geometric edits such as duplicating, resizing, and translating. Additionally, the motion of these elements can be adjusted by setting different motion attributes.

**Appearance Editing** Users can edit the appearance in a specific frame by drawing, stylizing, or recoloring, and these edits will be propagated across the entire video with cross-frame consistency. In

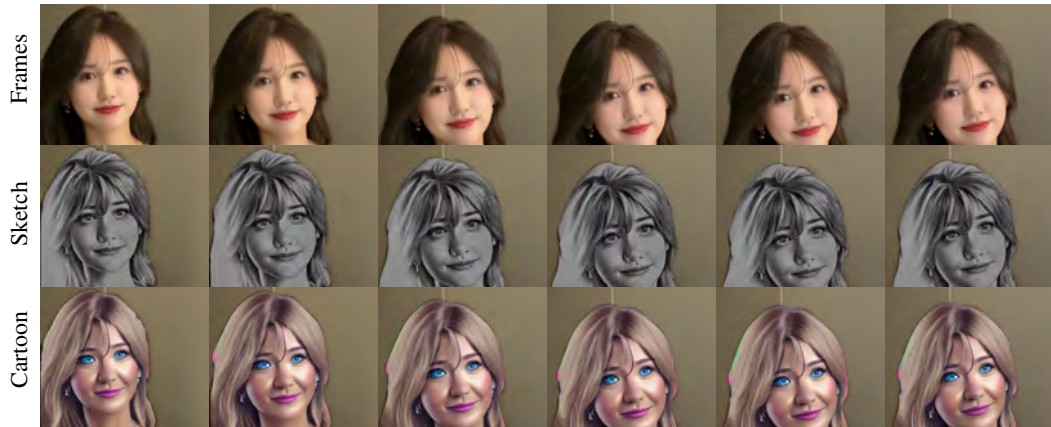

Figure 6: Appearance editing results using the 2D prompt editing method [60].

Fig.6, we demonstrate appearance editing using ControlNet[60]. Appearance editing is user-friendly in our representation, as it only requires single-frame editing.

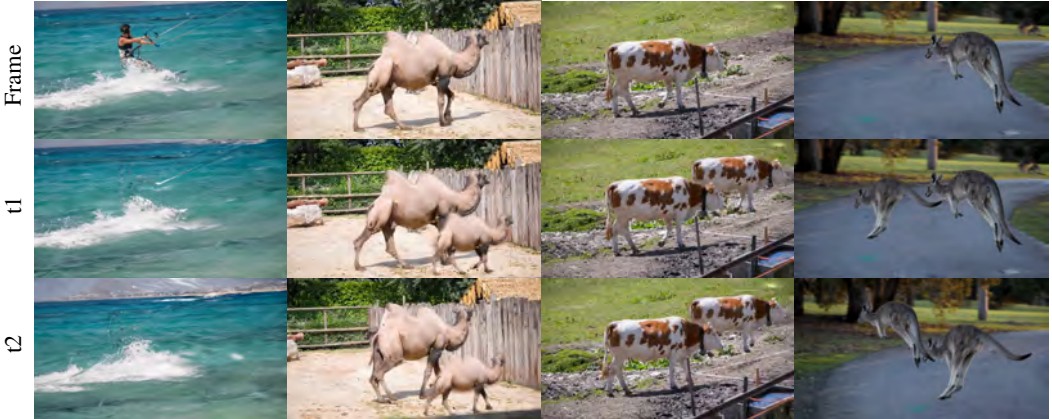

Figure 7: Geometry editing results including object deleting, resizing, copying, and translating.

**Novel View Synthesis & Stereoscopic Video Creation** Benefitting from depth regularization, the 3D Gaussians maintain a meaningful 3D structure, even from a monocular video. This facilitates novel view synthesis tasks, with examples provided in the supplemental video. Stereoscopic videos can also be produced, as shown in Fig. 8.

## 6.2 Ablation Study.

We perform ablation studies to validate the importance of the proposed modules, including camera model (perspective/orthographic), flow loss, depth loss (L2/scale-shift invariant). The results are reported in Table 2.

Table 2: Ablation of each module in our framework.

| Methods | Ours | Perspective Camera | w/o Flow Loss | w/o Depth Loss | L2 Depth Loss |
|---|---|---|---|---|---|
| PSNR ↑ | **29.61** | 22.51 | 25.16 | 29.18 | 28.15 |
| SSIM ↑ | **0.8624** | 0.6908 | 0.6937 | 0.8475 | 0.8214 |
| LPIPS ↓ | **0.1845** | 0.3958 | 0.4724 | 0.2449 | 0.3328 |

Using a predefined pinhole camera intrinsic led to unstable optimization, resulting in artifacts in both geometry and appearance. This instability likely stems from the rasterization process, where the denominators of Gaussians' screen coordinates UV include depth, complicating the gradients. Replacing the shift- and scale-invariant depth loss with absolute L2 loss degrades performance, as monocular depth cues are ambiguous regarding scale and shift.

Moreover, the *depth prior* is crucial for maintaining the 3D structure of Gaussians. Without the depth prior, Gaussians collapse into a flat 2D plane, hindering novel view synthesis and resembling 2D-layer methods.

We also demonstrate the significance of motion regularization (rigid loss) and the selection of motion coefficients n and l, which effectively suppress unorganized Gaussian motion. Please refer to Sec. A.3 for more details.

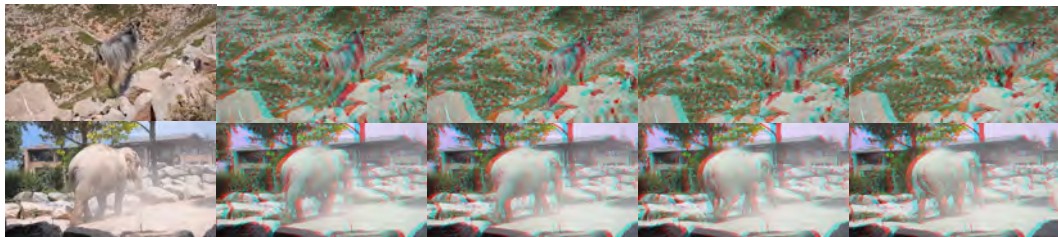

Figure 8: Stereo view synthesis. One original frame is visualized in the first column for comparison.

## 6.3 Limitations

Although achieving satisfying performance, there are still some limitations to be enhanced. First, our approach suffers from significant changes in the scene, since large deformation is hard to optimize. Initializing the scene with dynamic point clouds might alleviate this problem. In addition, our approach still relies on existing correspondence estimation methods (e.g., RAFT), which might fail when processing rapid and highly non-rigid motion. Extending this representation to more general scenarios is still worth exploring. We have illustrated two scenarios in Fig 9.

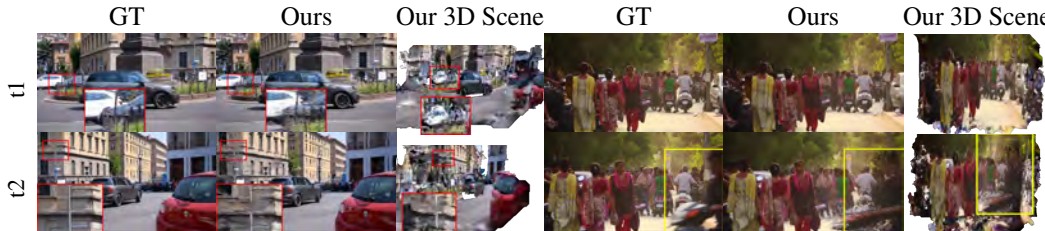

Figure 9: Our method may underperform in two scenarios. (1) Without camera estimation, it has to learn large camera rotations as scene motions, causing distant background blur (left). (2) It fails to track fast-moving transients, as photometric loss is insufficient for motion fitting (right).

## 7 Conclusion

In this paper, we introduced a novel explicit video Gaussian representation (VGR) based on 3D Gaussians to address the challenges of video processing. By modeling video appearance in a canonical 3D space and associating each Gaussian with time-dependent 3D motion attributes, our approach effectively handles complex motions and occlusions. Leveraging recent advancements in monocular priors, such as optical flow and depth, we lift 2D information into a compact 3D representation, facilitating a wide range of video-processing tasks. Our VGR method demonstrates efficacy in dense tracking, improving monocular 2D priors, video editing, interpolation, novel view synthesis, and stereoscopic video creation, providing a robust and versatile framework for sophisticated video processing applications.

## Acknowledgement

This work has been supported by Hong Kong Research Grant Council - Early Career Scheme (Grant No. 27209621), General Research Fund Scheme (Grant No. 17202422), and RGC Matching Fund Scheme (RMGS). Part of the described research work is conducted in the JC STEM Lab of Robotics for Soft Materials funded by The Hong Kong Jockey Club Charities Trust. We would like to thank Ziyi Yang for the insightful discussion and generous help.

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

# A  Appendix / Supplemental Material

## A.1  Implementation Details

Typically, we use a video clip of about 50-100 frames and train the system iteratively for 20,000 steps. The training duration is approximately 15-20 minutes on an NVIDIA 3090 GPU. The Gaussians are initialized as 10,0000 points randomly sampled in a $[-1, 1] \times [-1, 1] \times [0, 1]$ box. We use an orthographic camera for rendering for simplicity, which is fixed at the origin. We also modify the rasterization pipeline of 3DGS to support the orthographic projection by replacing the $J$ in EWA projection with $\begin{bmatrix} W/2 & 0 & 0 \\ 0 & H/2 & 0 \end{bmatrix}$, where $W$ and $H$ are the resolution of the image. For each attribute attached to Gaussians, we set different learning parameters and annealing strategies, list in Tab 3. Note that the dynamics of Gaussians' rotation is also modelled in the same way as position.

During training, the number of video Gaussians is adaptively adjusted as in vanilla Gaussian Splatting [16]. Every 100 steps, Gaussians with an accumulated gradient scale of positions above a threshold will be densified. Based on their projected size, they will be either split or cloned. Concurrently, Gaussians with opacities below a threshold will be pruned. To avoid floaters, the opacity of Gaussians is reset to 0.01 every 3000 steps. After optimization, there are around $10^5 - 10^6$ 3D Gaussian for a video containing $10^7 - 10^8$ pixels (resolution $\times$ frame number).

The loss weights for render, depth, flow, motion regularization, and label are set to $\lambda_{render} = 5.0$, $\lambda_{depth} = 1.0$, $\lambda_{flow} = 2.0$, $\lambda_{arap} = 0.1$, and $\lambda_{label} = 1.0$.

Table 3: Gaussian attributes table

| Attributes | Position | Rotation | Scaling | $\mathcal{SH}_0$ | $\mathcal{SH}_{1,2,3}$ | Polynomial | Fourier | Seg Label | SAM Feature |
|---|---|---|---|---|---|---|---|---|---|
| lr | 6e-5 | 1e-3 | 5e-3 | 2.5e-3 | 1.25e-4 | 1e-3 | 1e-3 | 1e-3 | 1e-3 |
| Annealed lr | 1.6e-6 | / | / | / | / | 1e-5 | 1e-5 | / | / |

## A.2  More Visual Comparison

We have visualized the comparison with 4DGS [51] and RoDynRF [23] in Fig 10. We also visualized the tracking comparison in Fig 11. Note that due to the inaccurate camera pose for the wild scene, the performance of 4DGS is very limited.

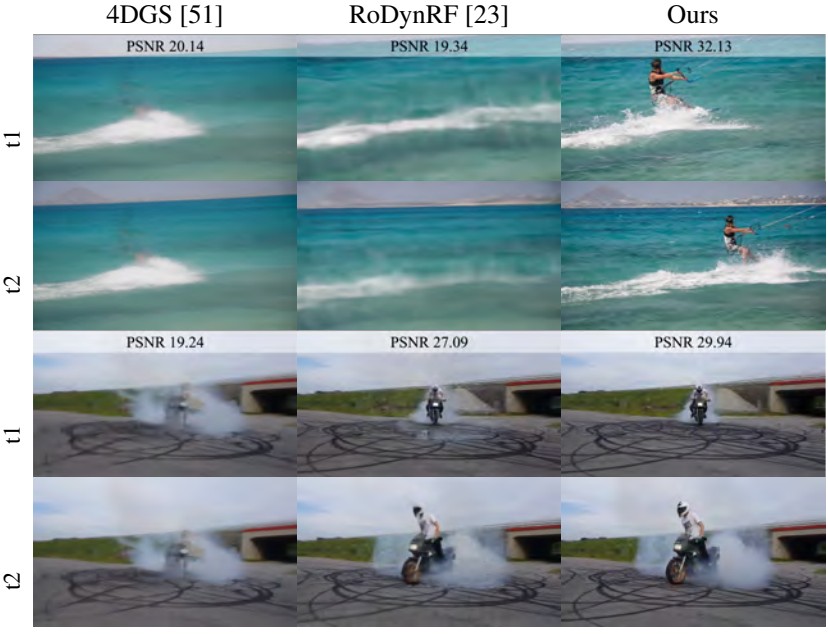

Figure 10: Video reconstruction compared with RoDynRF [23] and 4DGS [51]. Our method has higher PSNR and better visual quality.

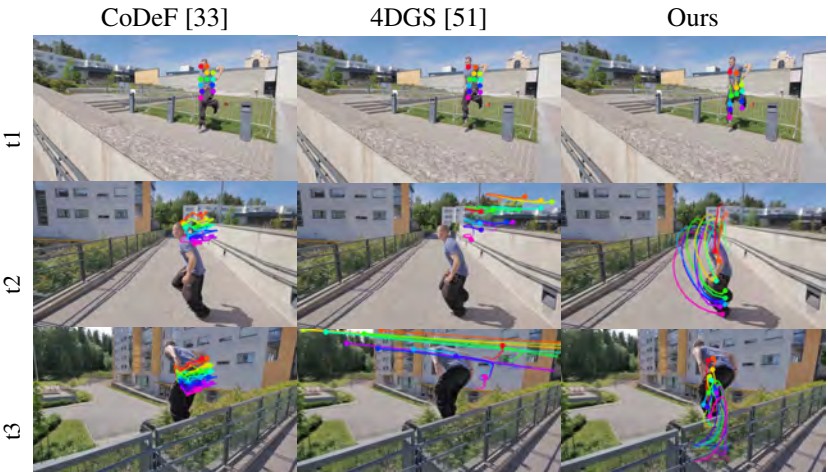

Figure 11: Tracking results visualization shows our method outperforms CoDeF [**?** ] and 4DGS [**?** ], especially in handling large-scale view changes and scene motions.

### A.3 Ablation Study

*Depth Regularization.* Without the depth prior, Gaussians collapse into a 2D plane. Although overfitting ability remains largely unaffected, the 3D structure is lost, and novel view synthesis is no longer possible, as shown in the right part of Fig. 12. Our approach then resembles 2D layer-based methods.

*Rigid Loss.* Without the rigid motion constraint, undesirable floaters appear, degrading rendering quality and reducing reconstruction PSNR by 1.51 dB, as illustrated in the left part of Fig. 12.

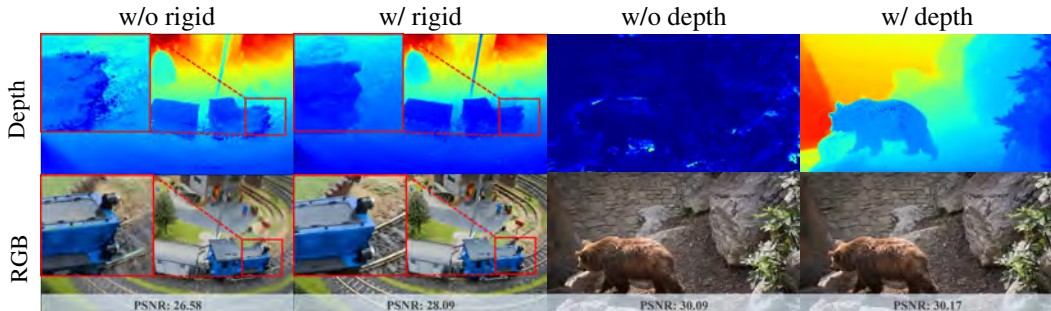

Figure 12: The depth prior (w/ depth) ensures video Gaussians conform to a realistic layout, while rigidity regularization (w/ rigid) eliminates floaters.

We also conducted ablation studies on the selection of camera models and depth loss formats as shown in Fig 13.

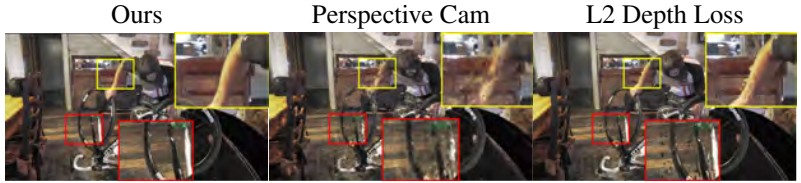

Figure 13: Ablation for camera model / L2 depth loss

We also evaluated the effect of different motion representation parameters (n/l in Eq 2), reported in Table 4 and Fig. 14.

Table 4: Ablation of n/l choice.

| Metric | n=8 / l=0 | n=0 / l=8 | n=l=4 | w/o n=l=8 | n=l=12 |
|---|---|---|---|---|---|
| PSNR ↑ | 28.02 | 26.87 | 28.46 | 29.61 | 27.47 |
| SSIM ↑ | 0.8392 | 0.7989 | 0.8512 | 0.8624 | 0.8357 |
| LPIPS ↓ | 0.3099 | 0.3245 | 0.2271 | 0.1845 | 0.2532 |

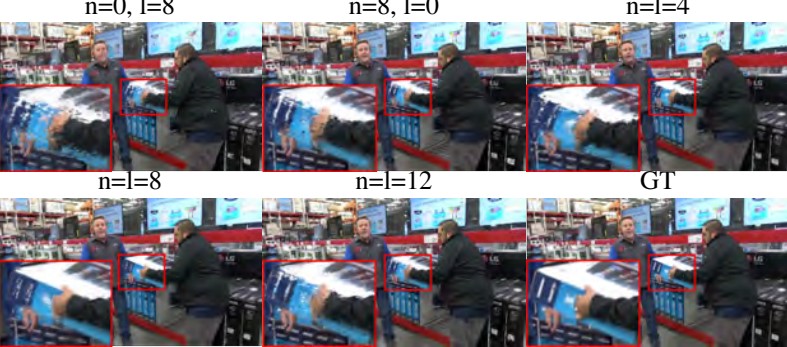

Figure 14: Ablation study on hyperparameters $n$ and $l$ reveals optimal quality at $n = l = 8$.

## A.4 Consistent SAM Feature Evaluation

By adding SAM features to Gaussian points, our method can obtain consistent SAM features. Compared with SAM extracted from per single frame, this feature has better consistency for moving objects and removes the requirements of passing through the SAM image encoder, hence obtaining more accurate video segmentation results at a faster speed, as reported in the Table 5 and Fig. 15.

Table 5: Comparision of per-frame SAM feature and ours.

| Metric | Per-frame SAM | Ours |
|---|---|---|
| IOU ↑ | 0.753 | 0.827 |
| Times ↑ | 0.513 | 0.025 |

## A.5 Video Interpolation

Thanks to the continuous parameterization of dynamics, our approach can interpolate video frames over time. We present the interpolation results in Fig. 16. Our method supports any video interpolation using an arbitrary continuous time re-mapping function at any frame rate.

## A.6 Multi-object Editing

By adding a multi-channel mask attribute to each Gaussian point, our method can achieve separate editing of multiple objects. We visualize an example of multi-object geometry / appearance editing in Fig. 17.

## A.7 Expanded Related Work

**Video Editing** Decomposing videos into layered representations facilitates advanced video editing techniques. Kasten et al.[14] introduced layered neural atlases that decompose an image into textured layers and learn a corresponding deformation field, thereby enabling efficient video propagation and editing. Subsequent advancements have introduced more sophisticated models. Ye et al.[58] developed deformable sprites, segregating videos into distinct motion groups, each driven by an MLP-based representation. Huang et al.[9] proposed employing a bi-directional warping field to support extensive video tracking and editing capabilities over longer durations. Recent innovations have also focused on enhancing the rendering of lighting and color details. Chan et al.[4] extended this approach by incorporating additional layers and introducing residual color maps, enhancing the representation of illumination effects within the video. The most current development in this

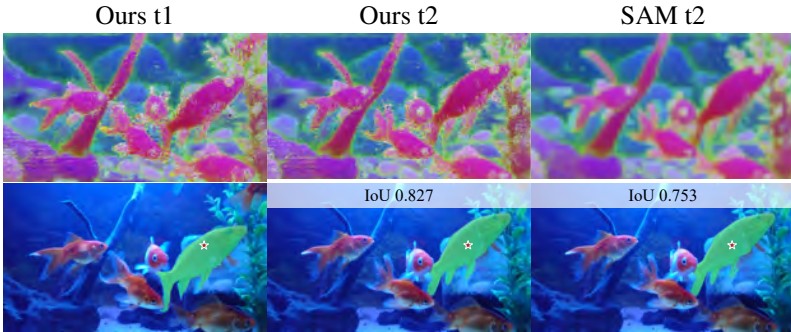

Figure 15: Segmentation comparison reveals that our lifted SAM features outperform per-frame SAM segmentation, delivering higher resolution feature maps and superior IoU.

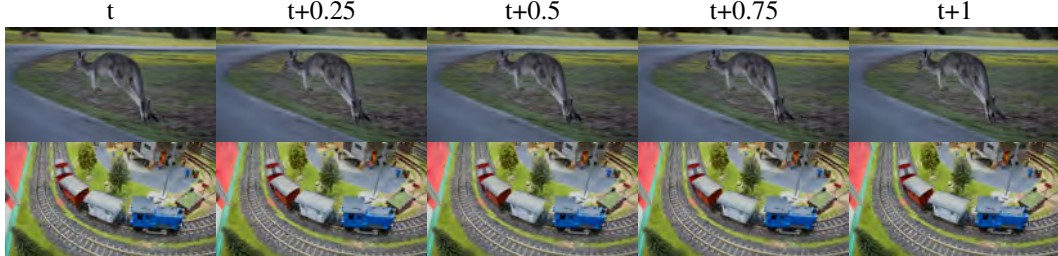

Figure 16: Video interpolation results. Please refer to the supplementary video for better visualization.

area is CoDeF [33], which leverages a multi-resolution hash grid and a shallow MLP to model frame-by-frame deformations relative to a canonical image. This approach allows for editing in the canonical space, with changes effectively propagated across the entire video. GenDeF [49] uses a similar representation to generate controllable videos.

Several studies have exploited the generative capabilities of latent diffusion models[38] for data-driven video editing. ControlVideo[61] adopts the methodology of ControlNet [60], integrating control signals into the network during the video reconstruction process to guide editing. Employing a related technique to manage control signals, MaskINT[28] utilizes frame interpolation to generate edited videos from specifically edited keyframes. In contrast, VidToMe[20] implements a token merging approach to incorporate control signals into the editing process. Additionally, certain research efforts [19, 62] have explored using inversion solutions to achieve video editing.

**Video Tracking.**    Video tracking is essential for capturing the physical motion of each point within a video sequence[39]. PIPs[8] track motion within fixed-size windows and include an occlusion branch, though they lack the ability to re-detect targets following prolonged occlusions. Building on the temporal processing concepts from PIPs, TAPIR [6] introduces TAP-Net[5], which precisely locates per-frame points. CoTracker[13] advances this by tracking individual query points using a sliding-window transformer approach. OminiMotion[48] pioneers the use of neural radiance fields[29] to model scene flow in NDC space. Its bijection network, which represents scene flow, is optimized for photometric consistency across frames, thereby enabling dense tracking. MFT[30] employs a sequential and dense point tracking methodology using optical flow fields computed across varying time spans. SpatialTracker[52] transforms each frame into a triplane and estimates trajectories by iteratively predicting movements with a transformer, facilitating 2D tracking within a 3D space. While state-of-the-art optical flow methods such as RAFT[44] and FlowFormer[11] provide accurate flow estimations for consecutive frames, they struggle with maintaining long-term frame correspondences.

**Gaussian Splatting**    Gaussian Splatting [16] has emerged as a potent method for enhancing rendering quality and speed in radiance fields. Following that, Lu et al. [25] further organize the Gaussians distribution by introducing anchor points, and Yang et al. [56] enrich its fitting ability in the specular setting with anisotropic sphere gaussian. These approaches have been extended to dynamic scenes in various recent studies. Luiten et al.[26] utilize frame-by-frame training, making it well-suited for

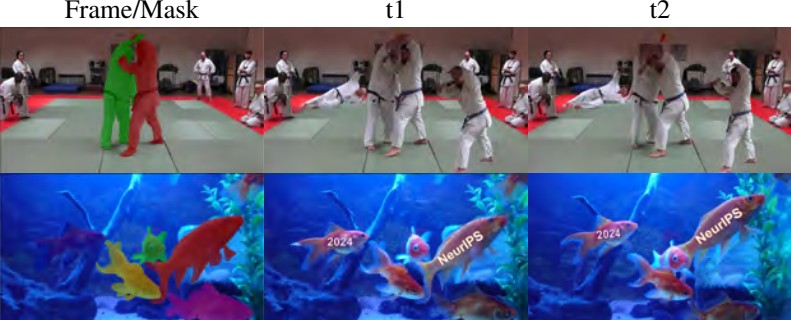

Figure 17: Segmentation and editing results are displayed from left to right: multi-object segmentation and video editing across two frames. The first row illustrates geometry editing where two individuals are copied and repositioned to lie on the ground and stand aside. The second shows the appearance editing with "2024" and "NeurIPS" painted on two fish.

multi-view scenes. Yang et al.[57] advance this by segmenting scenes into 3D Gaussians coupled with a deformation field, particularly for monocular scenes. Building upon this work, Wu et al.[51] have replaced the traditional MLP with multi-resolution hex-planes [3] and a shallow MLP. Additionally, Yang et al.[55] integrate time as an additional dimension in their 4D Gaussian model. SC-GS[10] introduces a novel approach using sparse control points to learn a spatially compact representation of scene dynamics. 3DGStream [43] offers a high-quality free viewpoint video (FVV) stream of dynamic scenes generated in real-time, though it necessitates multi-view video streams as input. Gaussian-Flow [22] hybrid the basis of polynomial and Fourier to represent the Gaussian motion. These methods typically rely on pre-estimated camera poses. Our approach specifically targets **monocular video representation**, obviating the need for camera pose estimations. This facilitates more robust long-term tracking and editing capabilities in dynamic scenes.

### A.8 Detailed Introduction to 3D Gaussian Splatting

Gaussian splatting [16] models 3D scenes using 3D Gaussians by learning from posed multiview images. Each Gaussian, denoted as $G$, is defined by a central point $\mu$ and a covariance matrix $\Sigma$,

$$G(x) = \exp\left(-\frac{1}{2}(x - \mu)^T \Sigma^{-1}(x - \mu)\right). \tag{14}$$

The covariance matrix $\Sigma$ undergoes decomposition into $RSS^T R^T$ for efficient optimization. Here, $R$ represents a rotation matrix, parameterized by a quaternion $q$ from $\mathbf{SO}(3)$, and $S$ is a scaling matrix defined by a positive 3D vector $s$. Additionally, each Gaussian is assigned an opacity value $\alpha$ to modulate its rendering impact and is equipped with spherical harmonic (SH) coefficients $sh$ for capturing view-dependent effects. The collection of Gaussians is represented as $\mathcal{G} = \{G_j : \mu_j, q_j, s_j, \alpha_j, sh_j\}$. Rendering is achieved through the equation:

$$C(u) = \sum_{i \in N} T_i \sigma_i \mathcal{SH}(sh_i, v_i), \text{ where } T_i = \Pi_{j=1}^{i-1}(1 - \sigma_j). \tag{15}$$

Here, $\mathcal{SH}$ denotes the spherical harmonic function and $v_i$ the viewing direction. The value of $\sigma_i$ is determined by evaluating the corresponding projection of Gaussian $G_i$ at pixel $u$ as follows:

$$\sigma_i = \alpha_i \exp(-\frac{1}{2}(u - \mu'_i)^T \Sigma'_i(u - \mu'_i)), \tag{16}$$

where $\mu'_i$ and $\Sigma'_i$ represent the projected 2D center and covariance matrix of Gaussian $G_i$, respectively. By optimizing the Gaussian parameters $\{G_j : \mu_j, q_j, s_j, \alpha_j, sh_j\}$ and dynamically adjusting Gaussian densities, high-quality and real-time image synthesis is facilitated. However, vanilla Gaussian splatting can only be used to represent a static scene. In this paper, we integrate this representation with video by assigning additional attributes to each Gaussian, enabling more versatile video processing.

