# OpenReview forum: "Splatter a Video: Video Gaussian Representation  for Versatile Processing"
_NeurIPS.cc/2024/Conference — NeurIPS 2024 poster_

### Official Review · Reviewer_rPzr · 2024-07-07

**Soundness:** 3
**Presentation:** 3
**Contribution:** 3
**Rating:** 7
**Confidence:** 4

**Summary:**

The work introduces a video Gaussian representation that leverages 2D priors, such as depth and optical flow, to regularize 3D Gaussians in for various 2D video tasks. This representation can be used for several downstream video processing and editing applications, including dense tracking, enhancing temporal consistency of 2D depth and features, geometry and appearance editing, video interpolation, novel video synthesis, and the creation of stereoscopic videos.

**Strengths:**

Unlike existing methods, this work learns Gaussian in a canonical 3D space (xyz) and reprojects them into the 2D domain as 2D frames for video tasks. This novel approach mitigates the negative impact of occlusions, facilitates the process of capturing complex motion, and enhances temporal consistency. To achieve these improvements, the work uses depth and optical flow to regularize the learning process. By incorporating label loss and non-rigid body constraints, the learned canonical 3D space supports several downstream tasks. The supplemental video clearly demonstrates the capabilities of the proposed representation.

**Weaknesses:**

- Learning one representation per clip (50-100 frames) takes 20 minutes, which is time-consuming. At 30 fps, encoding a one-minute clip requires 6 hours, and a ten-minute video takes approximately 2.5 days.

- The work can support several image features, such as SAM. However, different image features may require varying types and amounts of 3D Gaussians. For instance, RGB images need a greater number of smaller Gaussians, while segmentation requires fewer but larger Gaussians. The method may overestimate the number of required Gaussians.

- Although the method can enhance the temporal consistency for depth and SAM feature, it looks like the spatial domain is less smooth (more noisy) than these per-frame method in the video. Also, the per-frame method such as SAM can support different image per model, while the proposed method needs one model per clip.

- The performance of the method on depth estimation, segmentation mask estimation, tracking, novel view synthesis, and geometry editing in terms of quantitative results is unknown. The work only provides the PSNR of RGB reconstruction. Additionally, during geometry editing, human contours are still visible in the video.

- The model appears to be sensitive to the learning rate, requiring different learning rates for different losses (see Appendix Table 2).

- The work does not discuss failure cases. For instance, it does not explain why the PSNR for the elephant and cow are worse than the baselines. This could be due to hyperparameter issues. Also, how bad is it on drastic motion data?

**Questions:**

- What is the resolution and fps of the dataset?
- Compared to the baselines, does the proposed representation require more storage space?
- Is there a tradeoff between temporal smoothness and spatial details in the proposed method?
- How effective is the model on downstream tasks in terms of quantitative results?
- How difficult is it to tune the hyperparameters?
- To what extent can the novel synthesis work well?
- What is the inference speed? Does the unified framework slow down the inference speed of 3DGS?

**Limitations:**

I agree with the limitation mentioned by the author.
- One possible limitation is hard to capture long-duration clip or clip with long-range correspondence.

---

> ### Author Rebuttal · Authors · 2024-08-07
>
> We appreciate the reviewer's valuable feedback and address the concerns raised.
>
> **Training Speed**: While our method's training time relative to video duration is lengthy, it demonstrates significantly faster training speed compared to other methods, as shown in Table R1. We also report superior GPU memory and rendering efficiency (FPS). It's worth noting that once trained, these models can support various applications. Moreover, the speed may also potentially benefit from hardware advancements or parallel processing techniques.
>
> **Overestimation of Gaussians?**: This query touches on the different frequencies between RGB and feature images. While features with higher semantic content may skip low-level signals, our current approach didn't process these signals specially but focused on a universal video processing representation, which might lead to a over-parameterized Gaussians for features. Although a finer Gaussian representation might seem excessive, it prevents information loss and maintains sharp segmentation contours, ideal for handling the dense information in RGB images.
>
> **About SAM Feature**: The SAM feature appears smoother because it's interpolated from a lower-resolution tensor to full image resolution, whereas our rendering feature maintains the original resolution. Any high-frequency details smoothed during this process do not affect the final mask quality, as shown explicitly in Fig. R4 of the attached pdf file. Although SAM is more generalized, our method ensures better object integrity and segmentation quality across video frames, with more efficient processing time by skipping the image encoding step. The comparison is reported below:
>
> |  | Per-frame SAM | Ours |
> | --- | --- | --- |
> | IoU ↑ | 0.753 | 0.827 |
> | Time/s | 0.513 | 0.025 |
>
> **Sensitive to learning rate?**: All the videos share the same hyperparameters including loss weights and learning rate. We didn’t tune the parameters for different videos. Actually, most of the learning rates in Appendix Table 2 are derived from 3DGS.
>
> **Failure cases:**  We identify two scenarios where our method may underperform compared to dynamic GS methods or exhibit suboptimal performance, illustrated in Fig R7.
>
> -  The first scenario involves videos characterized by significant camera motion with few moving objects (e.g. elephant and cow in Table 1 of the paper). In these instances, the scene resembles a static environment where tools like COLMAP can effectively estimate precise camera poses. Dynamic GS methods model background motion using cameras extrinsic, whereas our method learns background trajectory during optimization. This approach imposes additional complexity on our training process, as our method must handle the bundle adjustment typically managed by COLMAP. In contrast, dynamic GS methods can fully use the camera prior, which is simpler for them to process.
> -  The second scenario involves videos with transient moving objects that appear only briefly, without continuous motion. In these cases, the estimated flow tends to be inaccurate, making it challenging for our method to learn pixel correspondence solely through the photometric consistency of Gaussian rendering.
>
> **Other Questions**
>
> **Resolution / FPS of dataset**:  We follow the Tap-Vid DAVIS benchmark (480p). The resolution of each video is 910 x 480 or 854 × 480, with a frame rate of 24 FPS.
>
> **Storage space**:  Our model size differs scene by scene, typically less than 100 MB, comparable with 4DGS, and much less than CoDeF (around 500M).
>
> **The trade-off between temporal smoothness & spatial details**:  Interesting observation. We feel there is no inherent trade-off between temporal smoothness and spatial details. This is due to the fact that natural motion patterns are often temporally smooth. Temporal smoothness helps prevent overfitting to the training data, avoiding fitting noise and thus it can even help improve spatial qualities.
>
> **Quantitative results on downstream tasks:** We have evaluated the tracking results on the Tap-Vid DAVIS benchmark. Our method outperforms most existing methods, except for Omnimotion. This is because Omnimotion is specifically designed for tracking tasks and performs poorly in terms of reconstruction and other metrics, and it does not support other editing tasks. In contrast, our method supports versatile video processing tasks, with higher computational efficiency and training efficiency. We also evaluate the segmentation results with our rasterized SAM feature on the gold-fish video, compared with SAM and reported in the above table.
>
> **Difficulty in tuning hyperparameters**:  We have not extensively fine-tuned hyperparameters such as learning rates. Our method shows robust performance and achieves the highest performance compared with existing methods with the same hyperparameters.
>
> **Extent of novel view synthesis**:  Our method effectively manages view changes in the application of generating stereoscopic videos. The specific boundary of view changes is determined by the accuracy of monocular estimation. With adequate depth supervision, our method can accommodate significant view warping. This important role of depth prior plays is also highlighted in the concurrent work "Shape of Motion: 4D Reconstruction from a Single Video."
>
> **Inference speed**: The inference speed of our method is a little slower than 3DGS but still fast enough for real-time applications, approaching over 150 FPS on a resolution of 854×480 on a single NVIDIA RTX3090 GPU.

---

> > ### Comment · Reviewer_rPzr · 2024-08-08
> > **Nice and extensive rebuttal**
> >
> > First of all, I would like to thank the authors for the extensive rebuttal and rMXd's detailed citation. The reviews are quite divergent, and it must be a tough time for the authors. Hence, I would like to provide my feedback earlier.
> >
> > I have read the rebuttal and other reviews. I am satisfied with the extensive rebuttal from the authors. I won't rate the novelty of the work too low even though the components of the framework are from existing work. The authors integrate these methods together on the VGR. There are no VGR-related works that have all these capabilities at the same time. Hence, the contribution for me is between fair to good.
> >
> > I also mentioned concerns about the evaluation on the downstream task. I am glad that the authors provided further evaluation on the Tap-Vid DAVIS benchmark and SAM in the rebuttal.
> >
> > Other issues, including hyperparameters, inference speed, failure cases, and model size, are also clarified by the authors.
> >
> > One weakness is the overestimation of Gaussian. However, since the model size is smaller than the baseline, I think it just fine.
> >
> > I decide to change my score from borderline accept to weak accept with detailed scores (3, 3, 2) at this time. I will keep following the discussion and may rate accordingly again. Thank all reviewers for the great reviews. I wish the authors good luck.

---

> > > ### Author Response · Authors · 2024-08-08
> > > **Response to Reviewer Feedback and Addressing Gaussian Overestimation Concerns**
> > >
> > > Thank you very much for your encouragement and support.
> > >
> > > We are pleased that our response could alleviate your concerns. This period has indeed been challenging for us, but we are also grateful because we have learned a lot from the reviewers' professional opinions and are trying to make the paper more systematic and comprehensive based on these suggestions.
> > >
> > > Regarding the issue of Gaussian overestimation, the number of Gaussians required by our method depends on the lowest level of supervision signal, which is RGB. Therefore, although there is some overestimation compared to some low-frequency high-level features, it does not introduce more Gaussians compared to other Gaussian-based methods such as 4DGS. Additionally, compared to other methods performing similar tasks like CoDeF, which relies on on large deformation model, our approach requires significantly fewer parameters.
> > >
> > > Although it is not the focus of our method, considering the different frequencies of various features comprehensively and designing a more efficient and compact Gaussian representation is a constructive suggestion and an interesting research direction. We will continue to explore this possibility in the future.
> > >
> > > Once again, thank you for your valuable time and advice!

---

> > ### Comment · Reviewer_rPzr · 2024-08-10
> > **More discussion!**
> >
> > Since there has been more discussion here, I would like to thank reviewers rMXd and 8X9e for further sharing their views. I would also like to share my personal view and my evaluation standards.
> >
> > Firstly, I have a few questions for the authors after reading reviewer rMXd's feedback. Although the answers to these questions may already exist in your rebuttal, I hope the authors can further verify whether my understanding is correct.
> >
> > 1. One of the differences between RoDynRF and your work is that RoDynRF models the real 3D world while your work models the pseudo 3D space, correct? If so, do you see this pseudo 3D space as a limitation or a novelty compared to RoDynRF? What do you think are the strengths and weaknesses of using pseudo 3D space?
> >
> > 2. Do you think the initial results you show represent the easiest case? If not, what is the most challenging part of this dataset? If I am correct, existing works do not outperform yours even on the easiest case, correct?
> >
> > 3. In Table R2, the method without depth loss is 0.43 lower than yours in terms of PSNR. Is this a marginal or significant performance difference in terms of the metric with the log term? Do you observe a substantial visual difference between these two settings?
> >
> > Again, I thank all reviewers for their views. Since the review is quite divergent, I would like to share my **personal** evaluation standard, which typically follows the NeurIPS guideline for authors, reviewers, and ACs.
> >
> > According to the strength section of the NeurIPS guidelines, **"Is the work a novel combination of well-known techniques? (This can be valuable!)."** Therefore, I believe the novelty of this work is at least not poor and is valuable. According to the rating guidelines of NeurIPS, this does not warrant a strong reject for me. A strong reject would be for a paper with major technical flaws, poor evaluation, limited impact, poor reproducibility, and mostly unaddressed ethical considerations. However, I do not see any major technical flaws, poor evaluation, limited impact, poor reproducibility, or mostly unaddressed ethical considerations in this work, especially **after reviewing the authors' rebuttal.** Therefore, I personally rate it a weak accept (Technically solid, moderate-to-high impact paper, with no major concerns regarding evaluation, resources, reproducibility, or ethical considerations) and believe it can be shared with the NeurIPS community. I see that the authors have provided **specific results rather than just arguments in the rebuttal.** With these specific results, I choose to trust the authors can integrate them well into the paper. Hence, I will leave this revision decision to the AC.
> >
> > Again, I may change my score according to the discussion in the future.

---

> > > ### Author Response · Authors · 2024-08-11
> > >
> > > Thank you for your further discussion. We are very grateful for your encouragement. We are committed to including all comments in our main paper or supplementary file.  We will also release the codes.
> > >
> > > **Real 3D world & Pseudo 3D space.**  *In terms of representation space*, this is a major difference between our method and RoDynRF. This is aligned with the *different objectives*: the goal of our method is a general video representation,  rather than accurate dynamic scene reconstruction, which is the objective of RoDynRF. The difference in representation also leads to *different outcomes and robustness*. Our method supports a wide range of video processing tasks, whereas RoDynRF is primarily limited to novel view synthesis, which is not our focus. Furthermore, in terms of video reconstruction quality, our representation method also consistently outperforms RoDynRF, as shown in Table R1: PSNR: 28.63 (ours) vs. 24.79 (RoDynRF). More detailed comparisons can be referred to in our response to reviewer rMXD. Based on the above, we believe that our pseudo-3D space is a significant advantage in achieving our goal of supporting various processing tasks while maintaining robustness with casual videos.
> > >
> > > **The benefits of using a pseudo 3D space mainly include the following aspects:**
> > >
> > > - **Better Reconstruction Quality, Enhanced Robustness, and Generalization to Casual Videos, and Beneficial to Optimization**:
> > >
> > >     (a) Unlike dynamic NeRF/GS methods such as RoDynRF and 4DGS, which focus on 4D world reconstruction, our pseudo-3D space design and optimization objectives are specifically tailored to transform a casual dynamic video into a 3D-aware space that supports various processing tasks. By employing a fixed orthographic camera model and rectifying the EWA projection of Gaussians, our pseudo-3D space design allows us to bypass the need for accurately estimating camera poses and intrinsic parameters—a task that is highly ill-posed when dealing with casually captured dynamic videos. This approach enables us to deliver better reconstruction quality and enhances the robustness and generalization of our method to casual videos compared to 4D reconstruction methods requiring accurate camera poses, delivering much better reconstruction quality (also see Table R1 and our response to reviewer rMXD)
> > >
> > >     (b) Furthermore, under our orthographic projection assumption, the movement of Gaussian points in the xy-coordinate directions directly corresponds to the magnitude of optical flow, while the depth-related loss only affects the z-coordinate. This significantly simplifies the optimization process. As long as we can obtain inter-frame 2D optical flow and a reasonable monocular relative depth, our method can represent the video in 3D space while preserving a coherent 3D structure.
> > >
> > >     These can be seen in our comparison with 4DGS and RoDynRF in Table R1 and in our ablation experiments in Table R2.
> > >
> > > **Compared to 2D representation such as CoDeF,  this pesudo 3D representation also delivers the following merits:**
> > >
> > > - **Modeling Complex Motion and Handling Occlusion:**  It helps us model more complex motions and handle occlusions in the scene. This is demonstrated in our comparison with CoDeF in Fig. 3 of the main text, and also in the comparison with 2D video representation methods CoDeF and Deformable Sprites in Table R2.
> > > - **Supporting Novel View Synthesis &Stereoscopic Video Generation and Spatial-aware Geometry Editing:** Even though it is not a real 3D world, the pseudo-3D space still has a reasonable 3D structure. This allows us to perform novel view synthesis within a certain range, as shown in Figure 8 of the paper. Moreover, the 3D spatial structure enables our method to handle occlusions more effectively, achieving spatial-aware geometry editing results. For example, in the third column of Figure 7 in the paper, we can insert a cow in front of the background and behind another cow, ensuring the correctness of the occlusion relationships. These capabilities are not achievable with 2D-based video representation.
> > >
> > > As illustrated in the failure case in Fig. R7, the main challenge with the pseudo-3D space arises when dealing with significant camera movements, particularly rotations. The intense motion of background points relative to a rapidly rotating camera often complicates the modeling of such global rigid movements, necessitating the introduction of additional motion constraints. Another limitation is that the pseudo-3D space is less effective at supporting novel view synthesis when there are large changes in viewpoint.

---

> > > > ### Author Response · Authors · 2024-08-11
> > > >
> > > > **Regarding “easiest case”**:   We do not believe that our method only works in the "easiest cases" as mentioned by reviewer rMXD. In fact, our method has demonstrated superior performance on almost all videos in the Tap-Vid DAVIS test set, which is also demonstrated in Table R1. Of course, there are indeed some very challenging videos in this set, such as the second example in the failure case of Fig R7, where our method struggles to handle the sudden appearance of a fast-moving person in the frame. However, none of the methods we have tested and compared so far have been able to handle this example effectively either.  In our test results, RoDynRF achieved a PSNR of only 11.38 on this dataset, while our method achieved 24.24.
> > > >
> > > > **Depth loss impact**:  The primary role of depth loss in our method is to model complex motions such as occlusion and to regularize the 3D structure. This is crucial for representing frames with significant occlusion and is essential for downstream tasks like novel view synthesis.  We have visualized the impact of depth loss in Figure 9 of the supplementary material. When depth loss is absent, the 3D structural information deteriorates, making the representation more akin to a 2D model. This compromises the ability to model occluded areas, as our approach intends. However, since occluded areas may occupy only a small percentage of the total pixels, this impact might not be directly reflected in pixel-level evaluation metrics, yet it significantly affects the overall visual quality.  To further illustrate this, we will include additional visualizations in our supplementary file. Additionally, we provide a case study on a scene with heavy occlusions, specifically the "libby" case. As shown in the table below, we observed a significant drop in PSNR from 28.02 to 26.71 when depth loss was omitted. Furthermore, without the depth loss, the representation becomes less effective in handling applications that require 3D information, such as stereoscopic video synthesis.
> > > >
> > > >
> > > >
> > > > | libby case | PSNR ↑ | SSIM ↑ | LPIPS ↓ |
> > > > | --- | --- | --- | --- |
> > > > | w/ Depth Loss | 28.02 | 0.7863 | 0.2991 |
> > > > | w/o Depth Loss | 26.71 | 0.7170 | 0.3527 |
> > > >
> > > > I hope these answers can resolve your questions. Thank you again for your support! We are happy to have  further discussions if you have any further concerns.

---

> ### Comment · Reviewer_rPzr · 2024-08-13
>
> I have read the new discussion and appreciate the authors' verification. I am considering increasing my score from weak accept to accept, but I can't promise anything yet. Please give me some time to reconsider the contribution part.

---

> > ### Author Response · Authors · 2024-08-13
> >
> > Thank you very much for your support! We believe that our method proposes a video 3D representation that effectively handles complex motions such as occlusion without the need for highly ill-posed 4D reconstruction, and it can benefit a range of downstream tasks (for specific details, please refer to our response "Benefit to downstream tasks” to reviewer axg5).
> >
> > Our method aims to demonstrate that the consistency between video frames can be constructed in a more fundamentally descriptive 3D space, even without camera poses, by leveraging current foundation models for flow and relative depth estimation. To achieve this, we are the first to use 3D Gaussian Splats (3DGS) under orthographic projection as a 3D representation of video. This approach not only maintains the high-quality rendering effects of 3DGS but also significantly simplifies the optimization difficulty by decoupling the screen coordinates from the depth coordinates. Our experimental results also prove that our method provides a new perspective and an effective approach for general video processing tasks.

---

> ### Comment · Reviewer_rPzr · 2024-08-13
>
> After reconsideration, I have decided to increase my post-rebuttal score to accept. Just to remind you, my pre-rebuttal score was borderline accept. I believe the paper deserves to be shared at NeurIPS. While I trust that the authors have the ability to reorganize the paper with the new results, I will leave the revision issue to the AC.

---

> > ### Author Response · Authors · 2024-08-13
> >
> > Thanks again for your support and trust !  We are committed to incorporating all feedback into our final paper version. And we will also publish our code accordingly to the community to facilitate the progress in this area.

---

### Official Review · Reviewer_axg5 · 2024-07-09

**Soundness:** 3
**Presentation:** 3
**Contribution:** 1
**Rating:** 6
**Confidence:** 4

**Summary:**

The goal in this work is to represent a video with a set of 3D Gaussian primitives, rendered with the 3DGS pipeline. The approach is optimisation-based and outputs video-specific Gaussians following a pre-defined trajectory model — hybrid of a polynomial and a Fourier series. The optimisation involves the use of data-based priors, such as monocular depth and optical flow networks. While the breadth of the experiments is laudable, it is limited to a few qualitative examples. The only quantitative results are available for video reconstruction in terms of PSNR.

**Strengths:**

* The goal is quite ambitious — representing a video with 3D Gaussians to support a variety of downstream tasks, such as dense tracking, interpolation and video editing.
* The approach follows quite naturally from the goals and makes use of available pre-trained models to supervise motion and depth estimation.
* The scope of (albeit qualitative) experimentation is quite compelling; the quantiative results on novel view synthesis are very encouraging.

**Weaknesses:**

* Representing a video with a set of 3D Gaussians is a rather straightforward extension of the original 3DGS pipeline. The only novelty is the trajectory model (Eq. 2), which was also employed in concurrent work [22].
* The experiments are predominantly qualitative. While it is reasonable for some tasks, such as editing, there are established benchmarks for the others (e.g. point tracking [8]). The only quantitative result in Tab. 1 is not an entirely fair comparison, since prior work does not use the specific monocular depth model used here.
* The ablation study is quite limited. Indeed, there is little technical contribution to study, except perhaps for the trajectory model.

*Post-rebuttal comment:* I thank the authors for their comprehensive response. If the additional results provided in the rebuttal are integrated into the revision, it will be a nice and interesting work to share with the community.

**Questions:**

The object masks are assumed available (from SAM). Do the static points corresponding to the camera motion also follow the trajectory formulation (c.f. Eq. (2))? Are there any assumptions about the camera intrinsic parameters?

**Limitations:**

Sec. 6.3 outlines some limitations. I would be curious to see a discussion on scaling the approach, especially in terms of handling multi-object videos.

---

> ### Author Rebuttal · Authors · 2024-08-07
>
> Thank you for your valuable advice!  We’d like to emphasize that using 3D representations to model and process casually captured in the wild videos, different from dynamic NeRF/GS,  is a much ill-posed problem.  Here, we design a system that integrates representations and regularization: 1) using ***a fixed orthographic camera***, alleviating the cumbersome camera pose estimation. 2) distilling monocular 2D priors for regularizing learning from such ill-posed scenarios with different regularizers from mono-depth and optical flow. We have illustrated some cases in which our model perfectly handles while dynamic NeRF/GS failed in Fig.R1 of the attached pdf file.
>
> Note that our approach doesn’t rely on the specific motion module. We adopt Fouier and polynomial bases due to their good trade-off between fitting capability and complexity.   Indeed, we have experimented with other motion representations in our exploration, which also achieved good results. We will report more results and analysis on different motion representations for this task.
>
> Moreover,  the main purpose of our work is not for 4D reconstruction, the primary objective for GS, but rather to perform versatile video processing tasks in the pseudo-3D space, which is wildly demonstrated in the main paper and Fig. R5.
>
> **Predominantly qualitative experiments:**    As our primary intention is to show the versatility of the approach on a bunch of video processing tasks which mostly focus on visual appearance, we focus on qualitative comparisons.  Here, to more thoroughly evaluate our approach quantitatively, we add more experiments on the DAVIS test videos following the TAP-Vid benchmark(480p) and report both the reconstruction and tracking metric in Table R1.  It can be seen that our approach significantly outperforms other methods in reconstruction.
>
> **Unfair comparison**: CoDeF is a 2D image-based video representation, where the depth prior cannot be used. In terms of Omnimotion, we observed that adding our relative depth loss results in lower-quality final results, as detailed in the table below. This is due to the depth maps in Omnimotion do not correspond to physical depth, as stated in their own paper.
>
> Comparison with Omnimotion with Depth Prior on camel dataset.
>
> |  | Omnimotion | Omnimotion + Depth | Ours |
> | --- | --- | --- | --- |
> | PSNR↑ | 23.9850 | 23.6139 | **30.48** |
> | SSIM↑ | 0.7055 | 0.6879 | **0.9299** |
> | LPIPS ↓ | 0.36447734 | 0.37741673 | **0.0849** |
>
> **Inadequate ablation:**  We conducted additional ablation studies focusing on the selection of coefficients n and l. The results are detailed below and Fig.R6. Our contribution extends beyond trajectory representation, and we have added more ablation studies regarding the camera model selection, different types of depth loss in Table.R2 and Fig.R6.
>
> |  | n = 8 / l = 0 | n = 0 / l = 8 | n = l = 4 | n = l = 8 | n = l = 12 |
> | --- | --- | --- | --- | --- | --- |
> | PSNR ↑ | 28.02 | 26.87 | 28.46 | **29.61** | 27.47 |
> | SSIM ↑ | 0.8392 | 0.7989 | 0.8512 | **0.8624** | 0.8357 |
> | LPIPS ↓ | 0.3099 | 0.3245 | 0.2271 | **0.1845** | 0.2532 |
>
> We also conducted ablation studies on the selection of camera models and depth loss formats as shown in Table R2 and Fig.R3. Using a predefined pinhole camera intrinsic led to unstable optimization, resulting in artifacts in both geometry and appearance. This instability likely stems from the rasterization process, where the denominators of Gaussians’ screen coordinates UV include depth, complicating the gradients. Replacing the shift- and scale-invariant depth loss with absolute L2 loss degrades performance, as monocular depth cues are ambiguous regarding scale and shift.
>
> **About camera motion and intrinsic:**  Our method employs a stationary camera, effectively merging the intrinsic motion of points with their motion relative to the camera. Consequently, all points in the scene are in motion and adhere to the trajectory formulation described in Equation 2.  As stated in lines #458-461 of the paper, our method utilizes an orthographic projection camera. Orthographic projection decouples the screen coordinates of points from their depth coordinates, simplifying the optimization difficulty, as demonstrated in Fig. R6 and Table 2.
>
> **Multi-object Videos:** With multi-object mask labels, our approach can be extended directly by increasing the dimension of mask feature. We evaluated the reconstruction quality of our model under two different conditions: merging the masks of different objects into a single-channel mask and increasing the mask dimensions to fit multiple objects. We conducted evaluations in two different scenarios, judo, and goldfish, as shown in the table below. The results indicate that the outcomes are essentially consistent across both settings. Additionally, we visualized examples of multi-object scene editing in Fig. R5.
>
> |  | Judo |  |  | Golden-fish |  |  |
> | --- | --- | --- | --- | --- | --- | --- |
> |  | PSNR↑ | SSIM↑ | LPIPS ↓ | PSNR↑ | SSIM↑ | LPIPS ↓ |
> | Single Mask | 39.90 | 0.9706 | 0.088 | 33.17 | 0.9410 | 0.1431 |
> | Multi Masks | 39.47 | 0.9712 | 0.083 | 33.35 | 0.9412 | 0.1445 |

---

> > ### Comment · Reviewer_axg5 · 2024-08-10
> > **Discussion**
> >
> > I thank the authors for their response. I have read the reviews and have been following the ongoing discussion.
> >
> > I do not have any major concerns about the quality or clarity. While I am not too excited about the technical contribution (also not claimed by the authors), I recognise that this can be subjective and some readers may disagree.
> >
> > Nevertheless, I still miss the significance in the results.
> >
> > The assumption of an orthographic camera and the use of a monocular depth network would not result in accurate 3D reconstruction. Perhaps as a consequence of this, the approach does not provide indisputable advantages on any of the downstream tasks. Overall, the work has a very applied nature, and I yet fail to see a single strong axis of a scientific impact.
> >
> > Moreover, there are strong signs that the initial submission -- which is the basis of my recommendation -- was rushed. It contains only preliminary, predominantly qualitative results. While I appreciate the additional experiments in the rebuttal, I feel that integrating them into the paper would require a significant revision.
> >
> > "This work aims to provide new insights into 3D representations for videos" -- may I ask, what are these insights?

---

> > > ### Author Response · Authors · 2024-08-11
> > >
> > > Thank you for your response and efforts in helping improve our paper. We are committed to incoporating all analysis into our main paper or supplementary file. We are also committed to releasing our codes.
> > >
> > > In our rebuttal, we want to emphasize that, unlike dynamic NeRF/GS, our work is not for dynamic reconstruction, but exists as a video representation to convert a video into a pesudo-3D space to support various processing tasks within this space. This is our core starting point, but it does not mean that we have no technical contribution.
> > >
> > > Firstly, as an early exploration, our primary goal is to develop a new representation of videos that supports versatile processing tasks rather than 3D real world reconstruction. To achieve our new video representation, we introduce a new approach that utilizes 3D Gaussians to transform a video into a pseudo-3D representation space, enabling processing within this space. The effectiveness of our method has been demonstrated both quantitatively and qualitatively across a variety of tasks. To the best of our knowledge, ours is **the only approach capable of enabling all these tasks.** Compared to other methods with similar objectives, such as CoDeF, our approach not only achieves significant performance gains across all evaluated settings but also enables new applications.
> > >
> > > Secondly, we address the significant challenges inherent in our new video formulation, specifically the ill-posed nature of monocular video data for 3D-aware appearance and motion modeling, as well as the complexities of camera modeling with unknown intrinsic and extrinsic parameters. We tackle these challenges with new designs and insights from both representation space design and optimization regularization.
> > >
> > > - **Representation Space Design:** we propose to employ a fixed orthographic camera model and study rectifying the EWA projection[1] from perspective to orthographic during the rasterization of Gaussians.
> > >
> > >     This EWA projection process is formulated as $\Sigma' = JW\Sigma W^TJ^T$, where $J$ is the Jacobian matrix of the projective transformation. Typically,  in the perspective projection,
> > >
> > > $$
> > > \left( \begin{matrix} u \\\\ v \end{matrix} \right) =
> > > \left( \begin{matrix} f_x * x / z + c_x \\\\ f_y*y/z + c_y \end{matrix} \right)
> > > $$
> > >
> > > $$
> > > J = \frac{\partial (u, v)}{\partial (x,y,z)} = \left( \begin{matrix} f_x/z & 0 & -f_x*x/z^2  \\\\ 0 & f_y/z  & -f_y * y/z^2 \end{matrix} \right)
> > > $$
> > >
> > > While in our orthographic camera model, the EWA projection needs to be modified as
> > >
> > > $$
> > > \left( \begin{matrix} u \\\\ v \end{matrix} \right) =
> > > \left( \begin{matrix} f_x * x  + c_x \\\\ f_y*y + c_y \end{matrix} \right)
> > > $$
> > >
> > > $$
> > > J = \frac{\partial (u, v)}{\partial (x,y,z)} = \left( \begin{matrix} f_x & 0 & 0  \\\\ 0 & f_y  & 0 \end{matrix} \right)
> > > $$
> > >
> > > which corresponds to lines #460-461 of the main paper.
> > >
> > > This approach allows us to bypass the challenges associated with camera pose estimation and camera intrinsic optimization, which, to the best of our knowledge, **is the first of its kind in the context of Gaussian representations**. This formulation has led to significant improvements in our experimental results, as demonstrated in Table R2: PSNR: 29.61 (our camera space) vs. 22.51 (perspective camera) and Fig R3. Additionally, our approach has proven to be more robust and consistently outperforms methods designed for novel view synthesis that rely on explicit camera pose estimation and intrinsic modeling, as shown in Table R1: PSNR: 28.63 (ours) vs. 24.79 (RoDynRF).
> > >
> > > - **Optimization in an Ill-Posed Setting**: Our key insight is that monocular cues from foundation models can serve as strong priors to regularize the decoupling of motion and appearance modeling.  Thus, we propose distilling 2D priors, including optical flow and depth, to guide the learning process. Note that when orthographic projection is used, the optical flow can correspond linearly to the xy coordinates of the Gaussian points, which greatly reduces the difficulty of optimization; while depth only affects the z coordinates, which is crucial when precise depth supervision is unavailable and only relative depth loss is used. This distillation has proven effective in training our representations, with optical flow regularization being particularly critical for quantitative performance.
> > >
> > > For clarification on the design and benefits of our 3D space for video processing, please refer to our response to reviewer rPzr. Additionally, for "comparisons to real-world 3D," please see our response to reviewer rMXD. We believe these contributions are both well-founded and effective in advancing our goal of representing videos in a new pseudo-3D space that supports versatile video processing tasks.
> > >
> > > [1] EWA volume splatting, Visualization, 2001

---

> > > > ### Author Response · Authors · 2024-08-11
> > > >
> > > > **Benefit to downstream tasks**:
> > > >
> > > > Our method has advantages over downstream tasks in the following aspects:
> > > >
> > > > 1) **For tracking**: Since the motion is lifted to 3D space, our method can better model the complex motion , e.g. occlusion, compared to 2D video representation methods. This is reflected in the tracking metrics in Table R1 (AJ↑:  Ours 41.9, CoDeF 7.6, Deformable Sprites 20.6).
> > > >
> > > > 2) **For novel view synthesis**: Thanks to the depth regularization, our 3D video representation can maintain a reasonable spatial structure, which enables our method to perform novel view synthesis and stereoscopic video creation within a certain range, as shown in Fig. 8 of the paper.  This is beyond the capabilities of 2D video representation methods (CoDeF and Deformable Sprites), and is also beyond the capabilities of methods designed specifically for tracking, such as Omnimotion (PSNR: Ours 28.63, Omnimotion 24.11).
> > > >
> > > > 3) **For spatial-aware editing**: Since the spatial structure is modeled, our method can support some editings related to spatial occlusion relationships. As shown in Fig.7 of the article, our method can naturally insert new objects in front of/behind the object to produce the effect of being occluded naturally.
> > > >
> > > > 4) **For faster and more precise segmentation**:  By adding SAM features to Gaussian points, our method can obtain consistent SAM features, as shown in Fig. 5 in the paper. Compared with SAM extracted from per single frame, this feature has better consistency for moving objects and removes the requirements of passing through the SAM image encoder, hence obtain more accurate video segmentation results at a faster speed, as reported in the table below and Fig. R4.
> > > >
> > > > |  | Per-frame SAM | Ours |
> > > > | --- | --- | --- |
> > > > | IOU ↑ | 0.753 | 0.827 |
> > > > | Time/s | 0.513 | 0.025 |
> > > >
> > > > 5) **For multi-object editing**:  By adding a multi-channel mask attribute to each Gaussian point, our method can achieve separate editing of multiple objects. We visualize an example of multi-object geometry / appearance editing in Fig. R5.
> > > >
> > > > **Too many qualitative results, not well prepared**:  As the primary goal of our paper is to explore a new pseudo-3D representation for video processing, we have followed the approach of CoDeF, a 2D video representation work, to demonstrate our representation's ability to support versatile applications. Unlike dynamic NeRF/GS methods, there are no existing benchmarks for quantitative studies on these tasks. Therefore, we also followed CoDeF in conducting qualitative illustrations and comparisons.
> > > >
> > > > In response to the reviewers' requests during the rebuttal stage, we added numerous comparisons with dynamic NeRF/GS methods, given our use of Gaussian representation. However, it is important to note that we are addressing two different types of tasks with different goals. While these additional results demonstrate the effectiveness of our method, they should not be used as a basis for evaluating the original paper's shortcomings.
> > > >
> > > > We are grateful for the valuable feedback from the reviewers, which has significantly helped us improve our work. We are committed to incorporating all suggested revisions into the main paper or supplementary file and will release all our source code.

---

> > > > > ### Author Response · Authors · 2024-08-11
> > > > >
> > > > > **Insights for 3D video representation**:  Generally speaking, videos are represented as pixels in each frame separately, which is indeed a very redundant representation. Work such as CoDeF proposes to use the content field and deformation field to model the content and motion of videos respectively, which can effectively support a range of video processing tasks.  However, motions in the 3D space are more intrinsic and include more transformations that cannot be modeled in 2D, such as occlusion. Therefore, we studying representing a video in compact 3D space is necessary and meaningful.
> > > > >
> > > > > However, recovering the real 3D from causual videos for 4D reconstruction requires the accurate estimation of camera poses and instrinics which is also not necessary for our goal of video processing. Thus, we propose a pesudo 3D space by employing a fixed orthographic camera model and studying rectifying the EWA projection[1] from perspective to orthographic during the rasterization of Gaussians.  This approach allows us to bypass the need for accurate camera pose estimation, enhancing robustness and generalization to casual videos, while enabling a variety of video processing tasks, as demonstrated in our paper.
> > > > >
> > > > >  Additionally, using an orthographic projection camera simplifies the process, as the motion of Gaussian points in screen space can be easily matched with optical flow, and the z-coordinate of Gaussian points can be constrained by relative depth.
> > > > >
> > > > > For more details on this space and its benefits, please refer to our responses to reviewers rMXD and rPzr, as well as Table R2, which highlights the effectiveness of this pseudo-3D space for video reconstruction.
> > > > >
> > > > > Another key insight we have gained from this work is that monocular priors, such as optical flow and monocular depth, can significantly enhance 3D-aware representations and improve the outcomes of various video processing tasks when properly integrated into the learning framework, offering strong regularizations. Given the growing trend of foundation models, we anticipate that their integration into video processing pipelines to support a wide range of tasks will become increasingly prevalent.

---

### Official Review · Reviewer_8X9e · 2024-07-10

**Soundness:** 4
**Presentation:** 3
**Contribution:** 3
**Rating:** 6
**Confidence:** 4

**Summary:**

This paper introduces a novel explicit 3D representation for video processing using video Gaussians. This method embeds videos into 3D Gaussians to model video appearance and motion in a 3D canonical space. By leveraging 2D priors such as optical flow and depth estimation to regularize the learning of video Gaussians, the approach ensures consistency with real-world content. The paper demonstrates the efficacy of this representation in various video processing tasks, including dense tracking, consistent depth and feature prediction, geometry and appearance editing, frame interpolation, novel view synthesis, and stereoscopic video creation. This method effectively handles complex motions and occlusions, offering a robust and versatile framework for sophisticated video processing applications.

**Strengths:**

1. Utilizing dynamic 3DGS as video represenation is well-motivated and rarely explored before.
2. The paper is generally well-written and easy to follow.
3. The paper represents various video processing tasks to showcase  superiority the proposed VGR, which convinces me in the experiment   part.
4. The limitation section includes a reasonable and honest analysis of the shortcomings of the current VGR.

**Weaknesses:**

1. More 4DGS-based methods, e.g,,[1],[2] should be compared and discussed in experiment. Note that I am not asking the author to compare the concurrent work like Gflow [3] or Mosca [4] but at least the aboved works presented in CVPR 2024 and ICLR 2024.
2. More abalation about the VGR representation itself but not the 2D prior should be presented in the paper. For example, the design of dynamic attributes and hybrid bases of dynamic Gaussian positions.





[1] Wu, Guanjun, et al. "4d gaussian splatting for real-time dynamic scene rendering." Proceedings of the IEEE/CVF Conference on Computer Vision and Pattern Recognition. 2024.

[2] Yang, Zeyu, et al. "Real-time photorealistic dynamic scene representation and rendering with 4d gaussian splatting." ICLR 2024

[3] Wang, Shizun, et al. "GFlow: Recovering 4D World from Monocular Video." arXiv preprint arXiv:2405.18426 (2024).

[4] Lei, Jiahui, et al. "MoSca: Dynamic Gaussian Fusion from Casual Videos via 4D Motion Scaffolds." arXiv preprint arXiv:2405.17421 (2024).

**Questions:**

1. How do you design the loss weight for the 2D prior? Is it case by case hyper-parameters?

**Limitations:**

I believe the the authors adequately addressed the limitations. I believe this paper gives a good example and standard, especially in the experimental parts for the fields of combining Gaussian Splatting with monocular videos.

---

> ### Author Rebuttal · Authors · 2024-08-07
>
> Thank you so much for your support in our work! We are delighted to answer the concerns raised.
>
> **Compare with 4DGS based methods:** Thank you for your suggestions. We have incorporated a comparison with 4DGS [1] on Tap-Vid DAVIS benchmarks for reconstruction and tracking tasks respectively. The results are reported in Table R1 above.  We also visualize qualitative results in Fig. R1 and Fig.R2 of the attached pdf file. Note that due to the inaccurate camera pose for the wild scene, the performance of 4DGS is very limited.
>
> Regarding [2], as the method does not employ 'deformation-based' modeling but instead uses static Gaussians in 4D space, it can not be used to establish correspondences. We acknowledge both [3] and [4] as significant concurrent works and will include them in the related work section.
>
> **More ablation:** We have performed an ablation study on the design of VGR by varying the hyperparameters (n and l). It is important to note that when n=0 or l=0, our representation simplifies to Fourier-only and polynomial-only, respectively. We report the results below and visualize the comparison in Fig. R6
>
> |  | n = 8 / l = 0 | n = 0 / l = 8 | n = l = 4 | n = l = 8 | n = l = 12 |
> | --- | --- | --- | --- | --- | --- |
> | PSNR ↑ | 28.02 | 26.87 | 28.46 | **29.61** | 27.47 |
> | SSIM ↑ | 0.8392 | 0.7989 | 0.8512 | **0.8624** | 0.8357 |
> | LPIPS ↓ | 0.3099 | 0.3245 | 0.2271 | **0.1845** | 0.2532 |
>
>  Thank you for the suggestion and we will include a discussion on this topic to enhance our paper.
>
> **Loss weight design:** The loss weights for render, depth, flow, motion regularization, and label are set to $\lambda_{render}$ = 5.0,$\lambda_{depth}$ =1.0, $\lambda_{flow}$ = 2.0, $\lambda_{arap}$ =0.1, and $\lambda_{label}$ =1.0 respectively. We have also included more ablations regarding the camera model design and each loss in Table R2 and Fig. R3 of the attached pdf file.  We will add more experimental details in the final version.  All of our experiments are performed with the same hyperparameters, which demonstrate the robustness of our approach.

---

> > ### Comment · Reviewer_8X9e · 2024-08-09
> >
> > After reading the other reviews and the rebuttal, it appears that the authors addressed the weakness points of the paper appropriately. I agree with reviewer  rPzr with the novelty concern, that is even though the components of the framework are from existing work, it's a nice try in a new field which should be encouraged.  I decide to keep my score.

---

> > > ### Author Response · Authors · 2024-08-11
> > >
> > > Thanks a lot for your support and encouragement!

---

### Official Review · Reviewer_rMXd · 2024-07-12

**Soundness:** 2
**Presentation:** 2
**Contribution:** 1
**Rating:** 2
**Confidence:** 5

**Summary:**

This paper presents a "Video Gaussian Representation" (VGR) to explicitly model the dynamic 3D scene contained in a monocular video. The VGR employs 3D Gaussian splatting as the backend, associating each Gaussian with time-dependent motion attributes. 2D priors, such as depth, optical flow, and optionally segmentation masks, are used to regularize the VGR optimization. The resulting VGR can be used for various video processing tasks, as claimed by the authors.

**Strengths:**

1. The paper is well-structured and easy to follow.
2. The visual illustrations are straightforward and clear.
3. The authors explore numerous downstream video processing tasks, demonstrating the proposed method’s versatility to some extent.

**Weaknesses:**

### 1. Lack of Novelty:
The primary objectives and main components of the proposed method are stemmed from existing works:
- Modeling video representation for video processing tasks is introduced by **CoDeF [CVPR'24]**:
  > CoDeF: Content Deformation Fields for Temporally Consistent Video Processing
- Motion dynamics modeled by polynomials and Fourier series is presented by **Gaussian-Flow [CVPR'24]**:
  > Gaussian-Flow: 4D Reconstruction with Dynamic 3D Gaussian Particles
- Depth loss is adopted from **MiDaS [TPAMI'22]**:
  > Towards robust monocular depth estimation: Mixing datasets for zero-shot cross-dataset transfer
- 3D motion regularization can be found in numerous dynamic modeling works, including Dynamic **3D Gaussians [3DV'24]** and **SpatialTracker [CVPR'24]**:
  > Dynamic 3D Gaussians: Tracking by Persistent Dynamic View Synthesis

  > SpatialTracker: Tracking Any 2D Pixels in 3D Space

### 2. Unclear Method Description:
- The **optimization procedure is missing** in the manuscript, which is crucial for understanding the methodology. Details such as how video frames are sampled and optimized, whether they are processed per-frame, batch-wise, or globally, and the approach to densification in new frames or under-reconstructed areas within the same frame, are not provided.
- The settings for $n$ and $l$ in the polynomials and Fourier series are **not explained**. Additionally, an ablation study regarding these parameters should be included.

### 3. Poor Evaluation:
According to the authors' description in Line #255:
> "Our approach is evaluated based on two criteria: 1) reconstructed video quality and 2) downstream video processing tasks"
However:
1. The reconstructed video quality is **only evaluated on 8 videos** selected from the DAVIS dataset, not the entire dataset (or test set), and only based on PSNR metrics, without considering other important metrics like LPIPS.
2. The downstream video processing tasks are only presented as visual illustrations, with **no quantitative evaluation**. For example, tasks like dense tracking should be evaluated on a well-established benchmark like **TAP-Vid [NeurIPS'22]**.
    > TAP-Vid: A Benchmark for Tracking Any Point in a Video

### 4. Unvalidated Claim:
The authors claim in Lines #227 and #232 that 2D features are distilled to 3D and applied to tasks such as video segmentation and re-identification. However, **no experimental or theoretical evidence** is provided to demonstrate that the **rasterized features** can be successfully used for segmentation.

### 5. Insufficient Literature Review:
The discussion of related works in dynamic 3D scene modeling only includes methods based on 3DGS, while NeRF-based methods such as **NR-NeRF [ICCV'21]** and **RoDynRF [CVPR'23]** should also be considered:
> RoDynRF: Robust Dynamic Radiance Fields

> Non-Rigid Neural Radiance Fields: Reconstruction and Novel View Synthesis of a Dynamic Scene From Monocular Video

**Questions:**

1. Please address the concerns listed in the weakness section.
2. How does the training/optimization time and model size compare with other methods?
3. How are the losses weighted exactly?
4. Do all videos are optimized according to same hyper-parameters?

**Limitations:**

Authors have discussed some limitations, but there are still other limitations that need to be addressed:
1. There are too many hyperparameters involved in the method. How do changes in these hyperparameters influence the results? Is the algorithm robust enough to effectively represent the video?
2. What are the potential failure cases? Some failure illustrations should be presented to provide readers with a better understanding.

---

> ### Author Rebuttal · Authors · 2024-08-07
>
> We thank the reviewer for your valuable time and the acknowledgment of our writing and illustrations. We are here to respond to the listed concerns.
>
> 1. **Novelty**:
>
> We would like to emphasize that our work is the first for dynamic GS without the camera dependency.  Our novelty doesn’t lie in any specific designs, but in the exploration to effectively lift casual videos from pixel to a more intrinsic 3D representation.
>
> - **Comparisons with CoDeF**: Our novelty lies in the exploration of representing **casual** videos by lifting to **3D** space, which is fundamentally different from CoDeF’s 2D representation.  To model a video,  CoDeF uses a canonical image and a network estimating the backward flow to model a video, which limits its flexibility in representing complex videos (Fig.3 of the main paper and Table.R1) and is incapable of video processing tasks requiring 3D like novel view synthesis and stereoscopic video creation.
> - **Comparison with 4D-GS methods like Gaussian-Flow**: Our key innovation is converting casual video pixels into intrinsic 3D representations without relying on camera pose estimation, which often fails in casually captured videos and is unachievable by current dynamic GS methods (refer to Fig.R1 and Table R1). Our approach can incorporate various motion-driving methods, though we primarily use Fourier and polynomial bases for their optimal balance of complexity and fitting ability, as discussed in our paper (Lines #135-143).
> - **Depth loss & 3D Motion Regularization**: For depth loss, the purposes and effects of the referred tasks and ours differ significantly. Our primary insight is to use monocular 2D clues to regularize the learning of 3D representations from videos, rather than to enhance depth estimation as in MiDaS.  In terms of motion regularization, we have cited related papers in lines #187-189 and did not emphasize this as a contribution. In summary, these losses themselves are not the focus; rather, it is the insights gained from employing such regularizers for learning in this ill-posed task that is crucial.
> 2. **Method Description**
> - **Optimization procedure missing**: Basically, our optimization procedure includes the following steps:  1) randomly sample one frame from the video sequence in each training iteration to construct the loss (Equation 9),  2)  the densification of Gaussians follows the same strategy as 3DGS, as stated in the line #212 of the main paper. We will release the code to the community for reproduction.
> - **Unexplained n/l & ablation**: We chose n=l=8 in our experiments based on the ablation of varying combinations, as shown below. Please refer to the Fig.R6 of pdf for visual comparison.
>
> |  | n = 8 / l = 0 | n = 0 / l = 8 | n = l = 4 | n = l = 8 | n = l = 12 |
> | --- | --- | --- | --- | --- | --- |
> | PSNR ↑ | 28.02 | 26.87 | 28.46 | **29.61** | 27.47 |
> | SSIM ↑ | 0.8392 | 0.7989 | 0.8512 | **0.8624** | 0.8357 |
> | LPIPS ↓ | 0.3099 | 0.3245 | 0.2271 | **0.1845**| 0.2532 |
> 3. **Evaluation**
>
> - **Insufficient Evaluation & Quantitative Evaluation on Dense Tracking**: Thank you for your feedback. Additional experiments on  the TAP-Vid DAVIS benchmark (480p) have been included, with both reconstruction and tracking metrics presented in Table.R1.
>
>     Despite Omnimotion's specialization in tracking, our approach supports a wider array of video processing tasks with higher computational and training efficiencies. It achieves comparable results with better reconstruction quality using fewer resources. Tracking performance comparisons with similar-cost methods (CoDeF / 4DGS) are shown in Fig. R2, highlighting our superior outcomes.
>
> 4. **Claim**
>
> Using the rasterized SAM feature with the SAM decoder has proven effective in image segmentation, as shown in Fig.R4. Our method ensures higher object integrity and also reduces the time needed for SAM encoding. The IoU and segmentation times, tested on an NVIDIA 3090GPU, are reported below.
>
> |  | Per-frame SAM | Ours |
> | --- | --- | --- |
> | IoU ↑ | 0.753 |0.827 |
> | Time/s | 0.513 | 0.025 |
>
> 5. **Literature**
>
> Thank you for your kind reminder. We will recognize the pioneering contributions of the paper "Non-Rigid Neural Radiance Fields" and the SOTA NeRF-based method RoDynRF. Additionally, we will include comparison results with RoDynRF in our analysis.
>
> 6. **Other Questions**
>
> - **Training time & Model size**: The training time is approximately 30 minutes, which is comparable to image-based methods, slightly shorter than 4DGS (about 40 minutes), and significantly less than NeRF-based methods (Omnimotion, RoDyF). We have also detailed the training times in the table above. The model size differs scene by scene, typically less than 100 MB, comparable with 4DGS and much less than CoDeF (500M).
>
> - **Loss weight**: The loss weights for render, depth, flow, motion regularization, and label are set to $\lambda_{render}$ = 5.0, $\lambda_{depth}$ =1.0,$ \lambda_{flow}$ = 2.0, $\lambda_{arap}$ =0.1, and $\lambda_{label}$ =1.0 respectively. We also ablate the importance of each loss and module of our method in Table R2.  We will add more experimental details in the final version.
>
> - **Optimization hyperparameters**: All of our experiments are performed with the same hyperparameters.
>
> 7. **Limitations**
>
> - **Hyperparameters impact**:
>     1) The impact of hyperparameters of the deformation model (n/l) has been evaluated in the Table above and Fig.R6.
>
>     2) As for the hyperparameters of learning rate in Table 2 of the paper, most of them are derived from 3DGS. The learning rate of newly added features(dynamic, mask) is referred to original Gaussian features.
>
>     3) The effect of each loss is also evaluated in Table R2 and Fig. R3
>
>     All of our experiments are performed with the same hyperparameters.
>
> - **Potential failure cases**:  We have demonstrated some failure cases in Fig. R7. Our approach fails to handle large motions, like intense background rotation or objects that suddenly appear.

---

> > ### Comment · Reviewer_rMXd · 2024-08-09
> >
> > Thank authors for the extensive rebuttal. I have read it as well as the other reviews. I also appreciate the contributions of reviewers rPzr and 8X9e for initiating the discussion and sharing their opinions. While I appreciate the authors' efforts, unfortunately, many of the concerns raised in my initial review remain unaddressed, and new concerns have arisen from the authors' feedback:
> >
> > ### **Major Concerns About Novelty**
> > Both reviewer axg5 and I initially expressed concerns about the novelty of this paper. I find it hard to believe that the composition of existing techniques can be considered novel, especially in a top-tier venue such as NeurIPS. At least one novel contribution should be presented in technical aspects, but this paper lacks such novelty, as I mentioned earlier.
> >
> > According to the authors’ response, they admitted that the paper does not offer technical novelty and instead claimed novelty in “converting casual video into intrinsic 3D representations without camera pose.” However, this claim raises additional concerns:
> >
> > 1. Several works, such as RoDynRF, have **already explored** using 3D representations to represent casual videos without camera pose.
> > 2. The concept of **"intrinsic 3D" seems overstated**. While we acknowledge that the authors apply 3DGS in this domain, it is a straightforward extension. The authors, however, disregard the powerful capability of 3DGS to model the real 3D world and instead use a simplified orthographic camera model to construct a **pseudo 3D space**. This approach does not represent a real physical 3D space and only considers the currently visible frame content, resulting the content outside the current frame is distorted. Both the supplementary video (00:03~00:11) and the figures (Fig. R7) illustrate these flaws. At least prior works like RoDynRF manages to reconstruct the real 3D world. Furthermore, according to Table R2 provided by the authors, canceling depth loss almost did not affect the reconstruction results, which further implies that the depth dimension in their 3D is artificial. The Gaussian points are optimized only to appear in the current frame, omitting real 3D spatial relationships.
> >
> > ### **Still Unclear Method Description**
> > Despite the authors’ further explanations, it is still unclear how the optimization procedures are carried out, particularly since this is a key aspect of the method. The authors stated that they “randomly sample one frame from the video sequence in each training iteration,” but according to Equations (3) and (5), the loss function must take two frames from different timesteps as input, which **contradicts the authors’ explanation**. Additionally, when a new Gaussian point is added at a certain timestep, how are the attributes of this Gaussian point initialized for all other timesteps? Reproducibility should begin with a clear method description, rather than relying solely on unpublished code.
> >
> > ### **Capability Concerns**
> > I appreciate the authors providing more results, especially the failure cases. With my extensive experience in processing videos, I noticed that the initial presentation in the paper only showed videos with nearly static backgrounds and clear foregrounds, which are the **easiest cases**. I was hoping the authors could demonstrate that the proposed method can handle more diverse videos. Unfortunately, the failure cases provided in the rebuttal suggest that even processing videos with simple camera rotations (e.g., car-roundabout) may fail, which echoes the flaws of the modeled fake 3D space. Thus, my concerns about the limited capability of the proposed method are confirmed.
> >
> > ### **Fairness Concerns**
> > I would like to thank the authors again for the extensive rebuttal. All reviewers, as well as the AC, acknowledge their efforts. However, this also reflects the fact that the initial submission was **largely incomplete**, particularly since no unselected quantitative results were presented in the paper. All four reviewers pointed out many missing quantitative experimental results, which are crucial for making this paper complete and solid. And the method description remains unclear even after the authors’ rebuttal, which further demonstrates that the paper was not well-prepared. I believe the necessary revisions will be so substantial that the paper will require re-review and should be resubmitted elsewhere. Otherwise, I would be very concerned about the fairness of the paper's acceptance, as it may suggest that all papers can be half-finished and completed during the rebuttal stage.
> >
> > **Therefore**, given the lack of technical novelty, the unclear method description, and the concerns regarding the capability and fairness of the paper, I maintain my score as a strong reject.
> >
> > ### *Follow-up Questions*
> > How were the results in Table R1 derived? Some methods require camera pose data, which the DAVIS dataset does not contain.

---

> > > ### Author Response · Authors · 2024-08-10
> > > **A More Detailed Response**
> > >
> > > Thank you for the further discussion to help improve our paper. We have carefully reviewed your feedback and believe that our work deserves a more fair evaluation.
> > >
> > > ## Concerns About Novelty
> > >
> > > We would like to further clarify that we do not acknowledge a lack of novelty or contributions. We believe that the key standard for judging the novelty of a piece of work should be whether it provides a new perspective for solving a problem. Even existing technologies, when analyzed from a new angle and capable of addressing failure cases that current methods cannot handle,  deserve recognition and encouragement.  Moreover, our work includes non-trivial designs that incorporate new insights, further showing its originality and value. Below, we will further elaborate on the points that we believe are valuable to the community.
> > >
> > > Firstly, as an early exploration, our primary goal is to develop a new video representation that supports versatile processing tasks, rather than focusing on 3D real-world reconstruction or exclusively on novel view synthesis like RoDynRF. We do not aim to achieve 3D dynamic real-world reconstruction in our paper and only mention that our approach can support novel view synthesis to some extent (see Lines 68-69), though this is not our primary focus. If the term “intrinsic 3D” gives the impression that our goal is 3D dynamic real-world reconstruction, we are willing to rephrase it as “3D-aware” to clarify our intent.
> > >
> > > To achieve our new video representation, we introduce a new approach that utilizes 3D Gaussians to transform a video into a pseudo-3D representation space, enabling processing within this space. The effectiveness of our method has been demonstrated both quantitatively and qualitatively across a variety of tasks. To the best of our knowledge, ours is the only approach capable of enabling all these tasks. Compared to other methods with similar objectives, such as CoDeF, our approach not only achieves significant performance gains across all evaluated settings but also enables new applications. We believe this fresh perspective and its promising results will contribute valuable insights to the advancement of video processing tasks within the community.
> > >
> > > Secondly, we address the significant challenges inherent in our new video formulation, specifically the ill-posed nature of monocular video data for 3D-aware appearance and motion modeling, as well as the complexities of camera modeling with unknown intrinsic and extrinsic parameters. We tackle these challenges with new designs and insights from both representation space design and optimization regularization.
> > >
> > > - **Representation Space Design:** we propose to employ a fixed orthographic camera model and study rectifying the EWA projection[1] from perspective to orthographic during the rasterization of Gaussians.
> > >
> > >     This EWA projection process is formulated as $\Sigma' = JW\Sigma W^TJ^T$, where $J$ is the Jacobian matrix of the projective transformation. Typically,  in the perspective projection,
> > >
> > >
> > > $$
> > > \left( \begin{matrix} u \\\\ v \end{matrix} \right) =
> > > \left( \begin{matrix} f_x * x / z + c_x \\\\ f_y*y/z + c_y \end{matrix} \right)
> > > $$
> > >
> > > $$
> > > J = \frac{\partial (u, v)}{\partial (x,y,z)} = \left( \begin{matrix} f_x/z & 0 & -f_x*x/z^2  \\\\ 0 & f_y/z  & -f_y * y/z^2 \end{matrix} \right)
> > > $$
> > >
> > > While in our orthographic camera model, the EWA projection needs to be modified as
> > >
> > > $$
> > > \left( \begin{matrix} u \\\\ v \end{matrix} \right) =
> > > \left( \begin{matrix} f_x * x  + c_x \\\\ f_y*y + c_y \end{matrix} \right)
> > > $$
> > >
> > > $$
> > > J = \frac{\partial (u, v)}{\partial (x,y,z)} = \left( \begin{matrix} f_x & 0 & 0  \\\\ 0 & f_y  & 0 \end{matrix} \right)
> > > $$
> > >
> > > which corresponds to lines #460-461 of the main paper.
> > >
> > > This approach allows us to bypass the challenges associated with camera pose estimation and camera intrinsic optimization, which, to the best of our knowledge, is the first of its kind in the context of Gaussian representations. This formulation has led to significant improvements in our experimental results, as demonstrated in Table R2: PSNR: 29.61 (our camera space) vs. 22.51 (perspective camera) and Fig R3. Additionally, our approach has proven to be more robust and consistently outperforms methods designed for novel view synthesis that rely on explicit camera pose estimation and intrinsic modeling, as shown in Table R1: PSNR: 28.63 (ours) vs. 24.79 (RoDynRF).

---

> > > > ### Author Response · Authors · 2024-08-10
> > > >
> > > > - **Optimization in an Ill-Posed Setting**: Our key insight is that monocular cues from foundation models can serve as strong priors to regularize the decoupling of motion and appearance modeling.  Thus, we propose distilling 2D priors, including optical flow and depth, to guide the learning process. Note that when orthographic projection is used, the optical flow can correspond linearly to the xy coordinates of the Gaussian points, which greatly reduces the difficulty of optimization; while depth only affects the z coordinates, which is crucial when precise depth supervision is unavailable and only relative depth loss is used. This distillation has proven effective in training our representations, with optical flow regularization being particularly critical for performance (See Table R2).
> > > >
> > > > We believe that these contributions are both well-founded and effective in advancing our goal of representing videos in a new pseudo-3D space that supports versatile video processing tasks.
> > > >
> > > > ---
> > > >
> > > > **Comparison with RoDynRF**:
> > > >
> > > > - **Different Objectives:**  We want to emphasize again that our goal is to explore a new explicit Gaussian-based pseudo-3D representation of videos to support a wide range of video processing tasks. In contrast, RoDynRF focuses on robust dynamic real-world reconstruction using an implicit representation  for novel view synthesis, with inaccurate camera intrinsic and extrinsic parameters. Therefore, the representations yielded by our approach are designed to support various applications, whereas the implicit representation from RoDynRF is primarily limited to novel view synthesis, as also demonstrated in their paper.
> > > > - **Different Representation Space Formulation:** RoDynRF still relies on explicit camera pose estimation and intrinsic parameter optimization, whereas our approach completely eliminates this dependency by employing a fixed orthographic camera model and rectifying Gaussian rasterization, as detailed in our response on novelty concerns (Representation Space Design). The explicit modeling of camera pose and intrinsic parameters makes RoDynRF less robust in handling casual videos, negatively impacting its performance, even though it is more capable than other existing methods that also depend on camera pose. This has been demonstrated by both the quantitative reconstruction quality of RoDynRF and the qualitative visualizations, where it fails in cases like “Kite-Surf” shown in Fig. R1. In contrast, our method discards the concept of camera pose entirely, allowing for more robust and consistent results.
> > > >
> > > > - **Different Outcomes:**  Given our different objectives, our representation, once learned, can support a wide range of video processing tasks, whereas RoDynRF is primarily limited to novel view synthesis, which is not our focus. Furthermore, our distinct formulation and representation result in consistently superior video reconstruction quality compared to RoDynRF, as shown in Table R1: PSNR: 28.63 (ours) vs. 24.79 (RoDynRF). Additionally, training RoDynRF requires significant computational resources, while our method is much more efficient. According to Table R1, processing a single scenario in the DAVIS dataset with RoDynRF takes over a day, whereas our method completes the task in just half an hour.
> > > >
> > > > Our goal is for the proposed representation to serve as a versatile interface for various video processing tasks. To achieve this, it must handle different types of videos robustly while preserving video quality. From this perspective, RoDynRF falls short. Its video reconstruction quality is already behind ours, and it lacks the capability to support multiple tasks.
> > > >
> > > > ---
> > > >
> > > > **Compare with Real-world 3D :**
> > > >
> > > > - As mentioned earlier, our primary goal is to develop a new video representation that supports versatile processing tasks, rather than focusing on 3D real-world reconstruction from casual videos—a task that requires precise modeling of camera intrinsic and extrinsic parameters, which, to our knowledge, is extremely challenging to achieve in casual videos. This difficulty is evident in the poor results we observed when reproducing existing 4D reconstruction methods, including 4DGS and RoDyF. Their video reconstruction quality is already unsatisfactory (see Table R1), let alone to achieve 4D real-world reconstruction.

---

> > > > > ### Author Response · Authors · 2024-08-10
> > > > >
> > > > > - Since our focus is on video processing tasks rather than 4D reconstructions, we introduce a pseudo-3D representation space that allows us to reconstruct videos at a much higher quality while supporting a variety of applications. To avoid confusion, we will rephrase “inherent 3D” as “3D-aware.” Compared to other approaches with similar objectives, such as CoDeF, our method excels at modeling complex motion, including occlusions, as demonstrated in Figure 3 of the main paper. Additionally, our pseudo-3D space enhances downstream tasks like novel view synthesis and stereoscopic video creation, producing reasonable results when viewpoint changes are not very large—capabilities that 2D video representation methods cannot achieve.
> > > > >
> > > > > - **Depth loss impact.** We have visualized the impact of depth loss in Figure 9 of the supplementary material. When depth loss is absent, the 3D structural information deteriorates, making the representation more akin to a 2D model. This compromises the ability to model occluded areas, as our approach intends. However, since occluded areas may occupy only a small percentage of the total pixels, this impact might not be directly reflected in pixel-level evaluation metrics, yet it significantly affects the overall visual quality. To further illustrate this, we will include additional visualizations in our supplementary file. Additionally, we provide a case study on a scene with heavy occlusions, specifically the "libby" case. As shown in the table below, we observed a significant drop in PSNR from 28.02 to 26.71 when depth loss was omitted. Furthermore, without the depth loss, the representation becomes less effective in handling applications that require 3D information, such as stereoscopic video synthesis.
> > > > >
> > > > >
> > > > > | libby case | PSNR ↑ | SSIM ↑ | LPIPS ↓ |
> > > > > | --- | --- | --- | --- |
> > > > > | w/ Depth Loss | 28.02 | 0.7863 | 0.2991 |
> > > > > | w/o Depth Loss | 26.71 | 0.7170 | 0.3527 |
> > > > >
> > > > > ---
> > > > >
> > > > > ## Unclear Method Description
> > > > >
> > > > > Thanks for asking for further clarification.  In our first response, our intention is to clarify the reviewer's question about the batch optimization process: “whether video frames are processed per frame, batch-wise, or globally.” Based on our understanding, the question asks how many video frames are processed during each forward/backward/optimizer step. Our response is that each batch processes only a randomly selected single frame at a single time step. Of course, when calculating the flow loss and ARAP loss, frames of other time steps are required. In practice, we randomly select another time step from the video frames to compute Equation (3) and Equation (5).  However, the render loss, depth loss and label loss are only calculated for the first frame.
> > > > >
> > > > > As we have responded, the densification strategy follows the same strategy as 3DGS.  Please note that in our method, all attributes of GS are time-independent, so there is no need to initialize them for each timestep separately. Following the densification strategy of 3DGS, whenever a new Gaussian point is added (either split or clone), it copies the corresponding attributes from the original Gaussian point, such as dynamic attributes or SAM features.
> > > > >
> > > > > We are more than willing to answer any questions regarding the code details and will open-source the code in the future to demonstrate the reproducibility of our method.
> > > > >
> > > > > ---
> > > > >
> > > > > ## Capability Concerns
> > > > >
> > > > > We have already reported our method's video representation capabilities in Table R1. The Tap-Vid DAVIS benchmark includes videos from various scenarios which can be checked from DAVIS official website , and in terms of video quality metrics (PSNR, SSIM, LPIPS), our method performs significantly better than other works. Here we report the metric of some specific cases, which includes more diverse motion patterns.
> > > > >
> > > > > |  | RoDynRF | CoDeF | Ours |
> > > > > | --- | --- | --- | --- |
> > > > > |  | PSNR / SSIM / LPIPS | PSNR / SSIM / LPIPS | PSNR / SSIM / LPIPS |
> > > > > | libby | 23.61 / 0.658 / 0.369 | 23.63 / 0.781 / 0.4 | 28.02 / 0.786 / 0.299 |
> > > > > | horsejump-high | 26.74 / 0.855 / 0.230 | 27.78 / 0.884 / 0.234 | 31.23 / 0.916 / 0.146 |
> > > > > | car-shadow | 26.5 / 0.838 / 0.337 | 25.41 / 0.837 / 0.316 | 35.42 / 0.947 / 0.151 |
> > > > > | car-roundabout | 20.48 / 0.605 / 0.518 | 23.98 / 0.804 / 0.321 | 24.85 / 0.8236 / 0.2429 |
> > > > >
> > > > > We have to recognize that our method does indeed have some limitations at present. In the “car-roundabout” case, although there is only camera rotation, the background motion is quite intense. Our method only decomposes the motion of Gaussian points into polynomial functions and Fourier series, so it currently struggles to handle such pronounced 3D rotations effectively. This is why it is categorized as a failure case. Even in this case, the “car-roundabout” scenario, our method's reconstruction metrics are superior to those of the dynamic NeRF method RoDynRF and the 2D video representation method CoDeF.

---

> > > > > > ### Author Response · Authors · 2024-08-10
> > > > > >
> > > > > > In the future, we will consider introducing explicit 3D rotations into the trajectory model to better handle this type of motion.
> > > > > >
> > > > > > ---
> > > > > >
> > > > > > ## **Fairness Concerns**
> > > > > >
> > > > > > Our work was originally intended as a new video representation. Our goal is not to perform 4D reconstruction like dynamic NeRF/GS, but rather to benefit various video processing tasks. The most closely related work to our method is CoDeF. In fact, most of CoDeF’s experimental results are presented quantitatively in their paper, so we followed a similar structure. During the rebuttal phase, we further enriched and refined the content of the paper based on the reviewers' suggestions. We are committed to incorporating all feedback into our main paper or supplementary material and will publish our code accordingly.
> > > > > >
> > > > > > Regarding the method description section, we responded according to the reviewers' questions. If there are any questions regarding the code details, please feel free to ask. We are more than willing to answer any detailed questions about the code and will release the codes.
> > > > > >
> > > > > > ---
> > > > > >
> > > > > > ## **Camera Estimation**
> > > > > >
> > > > > > In Table R1, only 4DGS relies on camera pose estimation. To address this, we first ran COLMAP on all the videos and successfully obtained camera poses for 14 out of 30 videos. For the remaining videos where COLMAP failed, we used UniDepth [2] to estimate consistent depth and then employed the DROID-SLAM [3] method to estimate the camera poses. While some videos may still contain significant errors in camera poses, there is currently no better way to handle these situations.
> > > > > >
> > > > > > It's important to note that, beyond COLMAP for pose estimation, we made substantial efforts to enhance the robustness of camera pose estimation by incorporating an advanced depth estimation approach. This is not included in the original 4DGS  paper. Despite these efforts, the performance remains unsatisfactory, further highlighting the challenges of accurate camera pose estimation in causal dynamic videos.
> > > > > >
> > > > > > [1] EWA volume splatting, Visualization, 2001
> > > > > >
> > > > > > [2] UniDepth: Universal monocular metric depth estimation, CVPR 2024
> > > > > >
> > > > > > [3] Droid-slam: Deep visual slam for monocular, stereo, and rgbd cameras, NeurIPS 2021

---

> > > > > > > ### Author Response · Authors · 2024-08-11
> > > > > > >
> > > > > > > We are happy to have further discussions if you have any further concerns.

---

> > > > > > > > ### Comment · Reviewer_rMXd · 2024-08-14
> > > > > > > > **Further Feedback**
> > > > > > > >
> > > > > > > > Thank authors for their response. I appreciate the efforts put into this work, but from a professional perspective, I maintain my current score for two main reasons:
> > > > > > > >
> > > > > > > > ### 1. Overstating the pseudo-3D space is disappointing.
> > > > > > > > The authors finally admitted that they are representing the video in a pseudo-3D space, and they claim that this representation benefits various applications. I fully understand the authors' explanation of the differences between pseudo-3D space and real-3D space, as well as their rationale for using pseudo-3D. However, based on the provided materials, it appears that only **simple videos** can be effectively represented to support video applications.
> > > > > > > > For example, in the supplementary video (00:28-00:32), the upper-right clip featuring a motorbike shows tracking results for only 17 frames out of a total of 42. It seems the latter frames were cut because the motorbike is occluded by foreground objects, such as flowers. I suspect that the proposed method fails to track the occluded object in these frames.
> > > > > > > > Furthermore, the presented videos all depict the easiest cases, which involve nearly static backgrounds and clear foregrounds, highlighting the limitations of a pseudo-3D world.
> > > > > > > > A method that can only handle simple videos and fails with slightly more complex ones is not what the community expects.
> > > > > > > > Prior works based on **real-3D space**, such as RoDyRF, have demonstrated their ability to **handle complex videos** in their teaser image, like "car-roundabout", where the proposed method fails. This capability is crucial for downstream video applications. Had the authors designed a method using real-3D space to support such applications, I would be far more excited. This is why I believe the authors are overstating the benefits of their pseudo-3D concept, leaving me disappointed.
> > > > > > > >
> > > > > > > > ### 2. The fairness about major revision.
> > > > > > > > Even if the method is based on pseudo-3D space, I would have considered a higher recommendation score if the initial submission had included sufficient experimental results. Attention, even one complete main results is acceptable. However, only cherry-picked quantitative results were presented, with many important and fair results missing. It is difficult for me to rate a paper that requires major revisions with a high score. This situation also raises concerns about fairness, which is a significant issue.
> > > > > > > >
> > > > > > > > ### Other comments
> > > > > > > > 1. I trust the overall results more, as it is possible to pick specific cases where the method performs better than others. If some cases are much better than other methods, then there must be cases where the results are not as good, or even worse, than other methods; otherwise, why are the overall results so close?
> > > > > > > > 2. Regarding the "motorbike" tracking example, one might infer that tracking could fail when dealing with occlusions. In that case, how were the three tracking metrics in Table R1 calculated? Since 2D tracking queries a 2D pixel coordinate that may contain multiple Gaussian primitives in 3D space.

---

### Author Rebuttal · Authors · 2024-08-07

We thank all the reviewers for their valuable comments. We are glad and appreciate that the Reviews for the comments of “well structured, easy to follow” (**rMXd, 8X9e**),  “novel and well motivated” (**8X9e, rPzr**), and recognize that the versatility video processing ability is  “well demonstrated, very ambitious and very encouraging” (**rMXd, 8X9e, axg5, rPzr**). We will release the code to the community to facilitate the progress in this area.

Below we first clarify our key novelty:  our work represents an early exploration of 3D-based representations of ***casually captured*** videos, enabling a variety of applications. Our aim is not to claim any specific designs as major innovations, but rather to highlight the system that effectively integrates different components to learn such representations from casually captured videos, despite the highly ill-posed nature of the problem. This work aims to provide new insights into 3D representations for videos.

We also greatly appreciate the reviewers' suggestions regarding the experimental section of our paper. In response, we have added experiments conducted on the **Tap-Vid DAVIS benchmark** (30 videos, 480p), providing a detailed report of various metrics. Additionally, we have included comparisons with dynamic NeRF/GS methods. It is noteworthy that in-the-wild videos often lack accurate camera poses, which significantly limits the performance of such methods.  The visual comparison is illustrated in the Fig.R1 of the attached PDF file.

Table R1.  Comprehensive comparison with existing methods on Tap-Vid benchmark (DAVIS).

| Metric | PSNR ↑ | SSIM ↑ | LPIPS ↓ | AJ ↑ | <$\delta_{avg}^x ↑$ | OA ↑ | TC ↓ | Training Time | GPU Memory | FPS |
| --- | --- | --- | --- | --- | --- | --- | --- | --- | --- | --- |
| 4DGS | 18.12 | 0.5735 | 0.5130 | 5.1 | 10.2 | 75.45 | 8.11 | ~40 mins | 10G | 145.8 |
| RoDyF | 24.79 | 0.723 | 0.394 | \ | \ | \ | \ | > 24 hours | 24G | < 0.01|
| Deformable Sprites | 22.83 | 0.6983 | 0.3014 | 20.6 | 32.9 | 69.7 | 2.07 | ~30 mins | 24G | 1.6 |
| Omnimotion | 24.11 | 0.7145 | 0.3713 | **51.7** | **67.5** | **85.3** | **0.74** | > 24 hours | 24G | < 0.01 |
| CoDeF | 26.17 | 0.8160 | 0.2905 | 7.6 | 13.7 | 78.0 | 7.56 | ~30 mins | 10G | 8.8 |
| Ours | **28.63** | **0.8373** | **0.2283** | 41.9 | 57.7 | 79.2 | 1.82 | ~30 mins | 10G | 149 |

Table R2.  Ablation of each module in our framework. We have ablated the motion regularization loss in the supplementary material. The ablation study of other modules is reported below, carried out on 5 videos ("bike-packing", "blackswan", "kite-surf", "loading", "gold-fish"), including diverse scenarios.

|  | Ours | Perspective Camera | w/o Flow Loss | w/o Depth Loss | L2 Depth Loss |
| --- | --- | --- | --- | --- | --- |
| PSNR ↑ | **29.61** | 22.51 | 25.16 | 29.18 | 28.15 |
| SSIM ↑ | **0.8624** | 0.6908 | 0.6937 | 0.8475 | 0.8214 |
| LPIPS ↓ | **0.1845** | 0.3958 | 0.4724 | 0.2449 | 0.3328 |

Please also pay attention to the attached PDF file for more visualization results.

---

### Decision · Program_Chairs · 2024-09-25

**Decision:**

Accept (poster)

**Comment:**

This paper received highly diverse ratings, where the discrepancy in the final rating boils down to a single question -- was the original submission too rough?

Extensive discussions were held between the AC, the SAC, and the reviewers, resulting in 64 forum replies (including reviews and the rebuttal), which far exceeds the typical number of posts for a single paper. All reviewers agreed that, in light of the rebuttal, the results seem solid, with the original claims in the paper holding. There was disagreement on whether the paper's presentation was clear enough, but this seems to have had mixed evaluations, making the AC believe that the quality was acceptable.

The contention was whether the original submission had enough results to support the claims made in the original submission. Two reviewers were in favour, as they were, although with qualitative highlights and only performed with a subset of the typical datasets used, demonstrated. One reviewer was initially negative but later changed their thoughts to conclude that it was enough, especially given that the original claims were supported by the rebuttal, and nothing changed concerning the claims. One reviewer remained negative, as the reviewer's opinion was that the original submission's results never had a "complete" set which would be required of a NeurIPS paper. An agreement was never met among the reviewers.

A significant concern raised by Reviewer rMXd was that the paper, according to the reviewer's point of view, was not finished at the time of submission, and accepting such work creates a precedent of unfair practice.  For example, an author could at the time of submission have the ideas written down, take the risk of not having the results, and then make them ready by the time of the rebuttal. This is clearly not what the NeurIPS FAQs suggest, as the FAQs state that the basis of the decision would be the original submission. The AC and the SAC also discussed whether this would be the case. It is both the AC and the SAC's opinion that **this is not the case.** The original submission, including the supplementary materials does provide at least a proof of concept result, which during the rebuttal phase was supported by numerical results. Within the reviewer discussions, it was also pointed out that NeurIPS does allow for an extra page for the camera ready, and also will have reviews public. It was the positive reviewers' consensus that these should be enough to have the revised version up to NeurIPS standards, without a major revision of the core of the paper, only adding the new experimental results provided in the rebuttal. To conclude, it is the AC's belief that the original submission had merit, which was further verified by the rebuttal, and that the original submission was perhaps incomplete **but not half-finished**.

Thus, the AC, in consultation with the SAC and extensive discussions with the reviewers, is recommending the paper for acceptance. The AC would further like to comment that the authors must include all of the results discussed in the rebuttal to the main paper.

The AC would further like to thank the extensive effort that the reviewers have taken to review the paper and participate in the discussions. The decision for this paper really boils down to philosophical debate and subjective evaluations of where we draw the line for a complete conference submission, especially related to empirical validation. The extensive debate that was held for this paper clearly demonstrates that this paper is on the borderline. While the current recommendation is to accept the paper, the AC would like to mention that either decision could have been justified. A critical component that was taken into account in the decision process was that the method remained identical, and the only things that were added were the numbers, including those on new data, and that the idea presented in the paper was interesting enough to excite reviewers to fight about it. Thus, sharing this with the broader research community would be valuable.